# Provably Efficient Offline Goal-Conditioned Reinforcement Learning with General Function Approximation and Single-Policy Concentrability

**Hanlin Zhu**
EECS, UC Berkeley; Meta AI
hanlinzhu@berkeley.edu

**Amy Zhang**
ECE, UT Austin; Meta AI
amy.zhang@austin.utexas.edu

## Abstract

Goal-conditioned reinforcement learning (GCRL) refers to learning general-purpose skills that aim to reach diverse goals. In particular, offline GCRL only requires purely pre-collected datasets to perform training tasks without additional interactions with the environment. Although offline GCRL has become increasingly prevalent and many previous works have demonstrated its empirical success, the theoretical understanding of efficient offline GCRL algorithms is not well established, especially when the state space is huge and the offline dataset only covers the policy we aim to learn. In this paper, we provide a rigorous theoretical analysis of an existing empirically successful offline GCRL algorithm. We prove that under slight modification, this algorithm enjoys an $\tilde{O}(\mathrm{poly}(1/\epsilon))$ sample complexity (where $\epsilon$ is the desired suboptimality of the learned policy) with general function approximation thanks to the property of (semi-)strong convexity of the objective functions. We only require nearly minimal assumptions on the dataset (single-policy concentrability) and the function class (realizability). Moreover, this algorithm consists of two uninterleaved optimization steps, which we refer to as $V$-learning and policy learning, and is computationally stable since it does not involve minimax optimization. We also empirically validate our theory by showing that the modified algorithm outperforms the previous algorithm in various real-world environments. To the best of our knowledge, this is the first algorithm that is both provably efficient with general function approximation and single-policy concentrability, and empirically successful without requiring solving minimax optimization problems.

## 1 Introduction

Goal-conditioned reinforcement learning (GCRL) aims to design agents that are able to learn general-purpose skills to reach diverse goals [Kaelbling, 1993, Schaul et al., 2015, Plappert et al., 2018]. In particular, offline GCRL learns goal-reaching policies by purely pre-collected data without any further interactions with the environment [Chebotar et al., 2021, Yang et al., 2022]. Since such interaction can be expensive or even unsafe in practice, offline GCRL is increasingly popular as a way to learn generalist agents in real-world environments [Lange et al., 2012, Levine et al., 2020].

Although offline GCRL is promising and achieves great success in various practical scenarios [Lynch et al., 2020, Chebotar et al., 2021, Yang et al., 2022, Ma et al., 2022b,c], designing practical algorithms that are provably efficient still remains an open question. On the practical side, an ideal algorithm should be scalable to huge (or infinite) state spaces and only require minimal dataset coverage assumptions. Moreover, the algorithm should be computationally efficient and stable (e.g., only

37th Conference on Neural Information Processing Systems (NeurIPS 2023).

using regression-based methods to train policies to avoid unstable minimax optimization). On the theoretical side, we aim to provide finite-sample guarantees of the learned policy.

Unfortunately, most existing algorithms are not both theoretically and practically efficient. On the one hand, many empirically efficient algorithms do not enjoy finite-sample guarantees [Lynch et al., 2020, Chebotar et al., 2021, Yang et al., 2022, Ma et al., 2022c] or even suffer constant suboptimality in favorable settings given infinite data (e.g., Ma et al. [2022c]). On the other hand, although many previous offline RL algorithms with theoretical finite-sample guarantees can be naturally extended to offline GCRL settings, they either cannot handle general value function approximation in the presence of huge (or infinite) state spaces [Jin et al., 2021, Rashidinejad et al., 2021, Yin et al., 2021, Shi et al., 2022, Li et al., 2022], or require impractically strong dataset coverage assumptions, such as all policy concentrability [Antos et al., 2008, Munos and Szepesvári, 2008, Xie and Jiang, 2021b].

Recently, several provably efficient algorithms have been proposed under general function approximation and single-policy concentrability [Zhan et al., 2022, Cheng et al., 2022, Rashidinejad et al., 2022]. In particular, the algorithm of [Zhan et al., 2022], based on the duality form of regularized linear programming formulation of RL, only requires the realizability assumption of function class. However, they all require solving minimax optimization problems which can be difficult or computationally unstable [Daskalakis et al., 2021]. On the contrary, some practically efficient algorithms (e.g., Ma et al. [2022b,c]) do not involve minimax optimization and thus are computationally more stable. This naturally raises an important question:

> *Can we design an efficient offline GCRL algorithm that enjoys favorable theoretical guarantees under mild assumptions and performs well empirically in real-world scenarios without a minimax formulation?*

In this paper, we answer the above question affirmatively by providing rigorous theoretical guarantees for an empirically successful offline GCRL algorithm named GoFAR proposed by Ma et al. [2022c]. We made some slight yet critical modifications to GoFAR. For deterministic MDPs, we need to carefully select the value of one hyperparameter that is set to 1 in the original GoFAR (which can be tuned in practice). For stochastic MDPs, we need to first learn the true transition model via maximum likelihood (MLE) and then plug in the learned model in the algorithm. To distinguish the difference between the original algorithm and the modified ones, we name the modified versions VP-learning (Algorithm 1).

We show that the VP-learning algorithm has both good empirical performance in real-world scenarios (already shown by Ma et al. [2022c], and we compare VP-learning and GoFAR empirically and show that our modification further improves the performance of the previous algorithm GoFAR) and favorable theoretical guarantees under mild assumptions. Specifically, it achieves $\tilde{O}(\text{poly}(1/\epsilon))$ sample complexity (where $\epsilon$ is the desired suboptimality level of the learned policy) under general function approximation with realizability-only assumption and partial data coverage with single-policy concentrability assumption. Moreover, the VP-learning algorithm can be decomposed into two uninterleaved learning (optimization) procedures (i.e., $V$-learning and policy learning), which only require solving regression problems without minimax optimization.

Note that the VP-learning algorithm can be naturally applied to single-task RL settings, and all the analysis in this paper does not rely on whether the setting is goal-conditioned. Since the original algorithm is proposed and empirically validated in goal-conditioned settings, we analyze the algorithm in goal-conditioned settings as well.

## 1.1 Related Work

Since our algorithm can be naturally applied to single-task offline RL settings, we discuss the related work in a broader scope, which also includes single-task offline RL.

**Offline RL in tabular and linear function approximation settings.** In tabular and linear settings, a line of work proposed efficient (both statistically and computationally) algorithms under single-policy concentrability [Jin et al., 2021, Rashidinejad et al., 2021, Yin et al., 2021, Shi et al., 2022, Li et al., 2022]. These algorithms construct uncertainty quantifiers to ensure pessimism such that policies not well covered by the dataset (which, by single-policy concentrability assumption, are thus suboptimal) suffer a large penalty. Yin and Wang [2021] also consider offline RL with single-policy

concentrability and achieve instance-dependent characterization. However, the above algorithms cannot be directly applied to many practical scenarios when non-linear function approximators are required, since it is hard to obtain uncertainty quantifiers without oracle access when function approximators are non-linear [Jiang and Huang, 2020, Jin et al., 2021, Uehara and Sun, 2021, Xie et al., 2021]. In our algorithm ($V$-learning step), we use a regularizer in the form of $f$-divergence instead of uncertainty quantifiers to ensure pessimism, which makes our algorithm efficient in the presence of non-linear function approximators without additional oracle access.

**Offline RL with all-policy concentrability.** Besides huge state spaces, another central challenge for offline RL is the lack of dataset coverability. Concentrability, defined as the ratio of occupancy frequency induced by a policy to the dataset distribution, is one of the most widely used definitions to characterize the dataset coverability [Munos, 2007, Scherrer, 2014]. Many previous works require all-policy concentrability to make the algorithm efficient [Szepesvári and Munos, 2005, Munos, 2007, Antos et al., 2007, 2008, Farahmand et al., 2010, Scherrer, 2014, Liu et al., 2019, Chen and Jiang, 2019, Jiang, 2019, Wang et al., 2019, Feng et al., 2019, Liao et al., 2020, Zhang et al., 2020, Uehara et al., 2020, Xie and Jiang, 2021a]. However, in practice, it is unreasonable to require that the offline dataset can cover all candidate policies, and our algorithm only requires single-policy concentrability.

**Offline RL with general function approximation and single-policy concentrability.** A recent line of work, which is based on marginalized importance sampling (MIS) formulation of RL, has shown success either empirically [Nachum et al., 2019a,b, Lee et al., 2021, Kim et al., 2021] or theoretically [Zhan et al., 2022, Rashidinejad et al., 2022]. In particular, Zhan et al. [2022], Rashidinejad et al. [2022] provide finite-sample guarantees for their algorithms under general function approximation and only single-policy concentrability. Another line of work [Xie et al., 2021, Cheng et al., 2022] also proposes provably efficient algorithms based on an actor-critic formulation under similar assumptions. However, all the above algorithms require solving minimax optimization, which could be difficult or computationally unstable [Daskalakis et al., 2021]. Instead, our algorithm only involves uninterleaved regression-based optimization without minimax optimization.

**Offline GCRL.** In the context of offline GCRL, the sparsity of the reward is another core challenge [Kaelbling, 1993, Schaul et al., 2015]. Several previous works aim to solve the issue and show empirical success without finite-sample guarantee [Ghosh et al., 2019, Chebotar et al., 2021, Yang et al., 2022, Ma et al., 2022c]. Although there exist theoretical studies of relevant problems such as the offline stochastic shortest path problem [Yin et al., 2022], theoretical understanding of the offline GCRL problem is still lacking. The most relevant work to this paper is Ma et al. [2022c], which shows great performance in several real-world settings without minimax optimization but lacks theoretical guarantee. In this paper, we proved that a slightly modified version of their algorithm (i.e., our VP-learning algorithm shown in Algorithm 1) is provably efficient. Also, we note that Ma et al. [2022a] can be viewed as the single-task version of Ma et al. [2022c], and our algorithm and analysis can be naturally extended to single-task offline RL settings.

## 2 Preliminaries

**Basic Notations.** Throughout this paper, we use $|\mathcal{X}|$ and $\Delta(\mathcal{X})$ to denote the cardinality and probability simplex of a given set $\mathcal{X}$. We use $x \lesssim y$ to denote that there exists a constant $c > 0$ such that $x \leq cy$, use $x \gtrsim y$ if $y \lesssim x$ and use $x \asymp y$ if $x \lesssim y$ and $y \lesssim x$. Also, we use the standard $O(\cdot)$ notation where $f(n) = O(g(n))$ if there exists $n_0, C > 0$ such that $|f(n)| \leq Cg(n)$ for all $n \geq n_0$, and denote $f(n) = \Omega(g(n))$ if $g(n) = O(f(n))$. For any $x \in \mathbb{R}$, define $x_+ \triangleq \max\{x, 0\}$; for any general function $f : \mathcal{X} \to \mathbb{R}$, define $f_+(x) = \max\{f(x), 0\}, \forall x \in \mathcal{X}$. Also, for any function $f : \mathbb{R} \to \mathbb{R}$, define $\bar{f}(x) = f(x) - \min_{u \in \mathbb{R}} f(u)$ for any $x \in \mathbb{R}$ if $\min_{u \in \mathbb{R}} f(u)$ exists and further overwrite the notation $\bar{f}_+(x) = \mathbb{1}\{f'(x) \geq 0\} \cdot \bar{f}(x)$ where $\mathbb{1}\{\cdot\}$ is the indicator function.

**Markov decision process.** We consider an infinite-horizon discounted Markov decision process (MDP), which is described by a tuple $M = (\mathcal{S}, \mathcal{A}, P, R, \rho, \gamma)$, where $\mathcal{S}$ and $\mathcal{A}$ denote the state and action spaces respectively, $P : \mathcal{S} \times \mathcal{A} \to \Delta(\mathcal{S})$ is the transition kernel, $R : \mathcal{S} \times \mathcal{A} \to \Delta([0, 1])$ encodes a family of reward distributions given state-action pairs with $r : \mathcal{S} \times \mathcal{A} \to [0, 1]$ as the expected reward function, $\rho : \mathcal{S} \to [0, 1]$ is the initial state distribution, and $\gamma \in [0, 1)$ is the discount factor. We assume $\mathcal{A}$ is finite while $\mathcal{S}$ could be arbitrarily complex (even continuous) as in many

real-world scenarios. A stationary (stochastic) policy $\pi : \mathcal{S} \to \Delta(\mathcal{A})$ outputs a distribution over action space for each state.

**Goal-conditioned reinforcement learning.**  In goal-conditioned RL, we additionally assume a goal set $\mathcal{G}$. Similar to Ma et al. [2022c], in goal-conditioned settings, the reward function $R(s; g)$ (as well as the expected reward function $r(s; g)$) and policy $\pi(a|s, g)$ also depend on the commanded goal $g \in \mathcal{G}$, and the reward no longer depends on the action $a$ and is deterministic.

Each (goal-conditioned) policy $\pi$ induces a (discounted) occupancy density over state-action pairs for any commanded goal $d^\pi : \mathcal{S} \times \mathcal{A} \times \mathcal{G} \to [0, 1]$ defined as $d^\pi(s, a; g) := (1 - \gamma) \sum_{t=0}^\infty \gamma^t \Pr(s_t = s, a_t = a; \pi)$, where $\Pr(s_t = s, a_t = a; \pi)$ denotes the visitation probability of state-action pair $(s, a)$ at step $t$, starting at $s_0 \sim \rho(\cdot)$ and following $\pi$ given commanded goal $g$. We also write $d^\pi(s; g) = \sum_{a \in \mathcal{A}} d^\pi(s, a; g)$ to denote the marginalized state occupancy. Let $p(g)$ be a distribution over desired goals, then we denote $d^\pi(s, a, g) = d^\pi(s, a; g)p(g)$ and $d^\pi(s, g) = d^\pi(s; g)p(g)$. An important property of occupancy density $d^\pi$ is that it satisfies the following Bellman flow constraint:

$$\sum_a d(s, a; g) = (1 - \gamma)\rho(s) + \gamma \sum_{s', a'} P(s|s', a')d(s', a'; g) \tag{1}$$

for all $s \in \mathcal{S}$ and $g \in \mathcal{G}$ when letting $d = d^\pi$ for any policy $\pi$. Moreover, any $d$ satisfying (1) is the occupancy density of a policy $\pi_d$ where

$$\pi_d(a|s, g) = \begin{cases} d(s, a; g)/d(s; g), & d(s; g) > 0 \\ 1/|\mathcal{A}|, & d(s; g) = 0 \end{cases} \quad \text{and} \quad d(s; g) = \sum_{a \in \mathcal{A}} d(s, a; g). \tag{2}$$

An important quantity associated with a policy $\pi$ is the value function, which is the expected discounted cumulative reward defined as $V^\pi(s; g) := \mathbb{E}\left[\sum_{t=0}^\infty \gamma^t r_t \mid s_0 = s, a_t \sim \pi(\cdot|s_t, g) \, \forall \, t \geq 0\right]$ starting at state $s \in \mathcal{S}$ with a commanded goal $g \in \mathcal{G}$ where $r_t = R(s_t; g) = r(s_t; g)$. We use the notation $J(\pi) := (1 - \gamma)\mathbb{E}_{s \sim \rho, g \sim p}[V^\pi(s; g)] = \mathbb{E}_{(s,a,g) \sim d^\pi}[r(s; g)]$ to represent a scalar summary of the performance of a policy $\pi$. We denote by $\pi^*$ the optimal policy that maximizes the above objective and use $V^* := V^{\pi^*}$ to denote the optimal value function.

**Offline GCRL.**  In this paper, we focus on offline GCRL, where the agent is only provided with a previously-collected *offline dataset* $\mathcal{D} = \{(s_i, a_i, r_i, s_i', g_i)\}_{i=1}^N$. Here, $r_i \sim R(s_i; g_i)$, $s_i' \sim P(\cdot \mid s_i, a_i)$, and we assume that $g_i$ are i.i.d. sampled from $p(\cdot)$ and it is common that data are collected by a behavior policy $\mu$ of which the discounted occupancy density is $d^\mu$. Therefore, we assume that $(s_i, a_i, g_i)$ are sampled i.i.d. from a distribution $\mu$ where $\mu(s, a, g) = p(g)d^\mu(s, a; g) = d^\mu(s, a, g)$. Note that we use $\mu$ to denote both the behavior policy and the dataset distribution. We also assume an additional dataset $\mathcal{D}_0 = \{(s_{0,i}, g_{0,i})\}_{i=1}^{N_0}$ where $s_{0,i}$ are i.i.d. sampled from $\rho(\cdot)$ and $g_{0,i}$ are i.i.d. sampled from $p(\cdot)$. The goal of offline RL is to learn a policy $\hat{\pi}$ using the offline dataset so as to minimize the sub-optimality compared to the optimal policy $\pi^*$, i.e., $J(\pi^*) - J(\hat{\pi})$, with high probability.

**Function approximation.**  To deal with huge state spaces, (general) function approximation is necessary for practical scenarios. In this paper, we assume access to two function classes: a value function class $\mathcal{V} \subseteq \{V : \mathcal{S} \times \mathcal{G} \to [0, V_{\max}]\}$ that models the value function of the (regularized) optimal policies, and a policy class $\Pi \subseteq \{\pi : \mathcal{S} \times \mathcal{G} \to \Delta(\mathcal{A})\}$ consisting of candidate policies. For stochastic MDP (Section 3.1.2), we also need a transition kernel class $\mathcal{P} \subseteq \{P : \mathcal{S} \times \mathcal{A} \to \Delta(\mathcal{S})\}$ which contains the ground-truth transition kernel. Additionally, for any function $f : \mathcal{S} \times \mathcal{G} \to \mathbb{R}$, we denote the operator $\mathcal{T} : \mathbb{R}^{\mathcal{S} \times \mathcal{G}} \to \mathbb{R}^{\mathcal{S} \times \mathcal{A} \times \mathcal{G}}$ as $(\mathcal{T}f)(s, a; g) = \mathbb{E}_{s' \sim P(\cdot|s,a)}[f(s'; g)]$. Also, for any function $V : \mathcal{S} \times \mathcal{G} \to [0, V_{\max}]$, we define $A_V(s, a; g) = r(s; g) + \gamma \mathcal{T}V(s, a; g) - V(s; g)$.

**Offline data coverage assumption.**  Our algorithm works within the single-policy concentrability framework [Rashidinejad et al., 2021], which is defined as below.

**Definition 1** (Single-policy concentrability for GCRL). *Given a policy $\pi$, define $C^\pi$ to be the smallest constant that satisfies $\frac{d^\pi(s,a,g)}{\mu(s,a,g)} \leq C^\pi$ for all $s \in \mathcal{S}$, $a \in \mathcal{A}$ and $g \in \mathcal{G}$.*

The single-policy concentrability parameter $C^\pi$ captures the coverage of policy $\pi$ in the offline data. Our algorithm only requires this parameter to be small for $\pi_\alpha^*$ which is a regularized optimal policy

(see Section 3 for formal definition) and is close to some optimal policy. This assumption is similar to Zhan et al. [2022] and and is much weaker than the widely used all-policy concentrability that assumes bounded $C^\pi$ for all $\pi$ (e.g., Scherrer [2014]).

## 3 Algorithms

Offline GCRL can be formulated as the following program:

$$\max_\pi \quad \mathbb{E}_{(s,g)\sim d^\pi(s,g)}[r(s;g)]. \tag{3}$$

(3) requires solving an optimization problem over the policy space. One can also optimize over occupancy density $d(s,a;g)$ s.t. $d = d^\pi$ for some policy $\pi$ which is equivalent to that $d$ satisfies Bellman flow constraint (1). Therefore, the program (3) can be represented equivalently as follows:

$$\max_{d(s,a;g)\geq 0} \quad \mathbb{E}_{(s,g)\sim d(s,g)}[r(s;g)]$$
$$\text{s.t.} \quad \sum_a d(s,a;g) = (1-\gamma)\rho(s) + \gamma \sum_{s',a'} P(s|s',a')d(s',a';g), \quad \forall(s,g) \in \mathcal{S} \times \mathcal{G}. \tag{4}$$

Let $d^*$ denote the optimal solution of (4), then (one of) the optimal policy can be induced by $\pi^* = \pi_{d^*}$ as in (2). Under partial data coverage assumptions, (4) might fail in empirical settings by choosing a highly suboptimal policy that is not well covered by the dataset with constant probability. Similar to Zhan et al. [2022], Ma et al. [2022c], a regularizer is needed to ensure that the learned policy is well covered by the dataset. Therefore, one should instead solve a regularized version of (4), which is stated as follows:

$$\max_{d(s,a;g)\geq 0} \quad \mathbb{E}_{(s,g)\sim d(s,g)}[r(s;g)] - \alpha D_f(d\|\mu)$$
$$\text{s.t.} \quad \sum_a d(s,a;g) = (1-\gamma)\rho(s) + \gamma \sum_{s',a'} P(s|s',a')d(s',a';g), \quad \forall(s,g) \in \mathcal{S} \times \mathcal{G}, \tag{5}$$

where the $f$-divergence is defined as $D_f(d\|\mu) \triangleq \mathbb{E}_{(s,a,g)\sim\mu}[f(d(s,a,g)/\mu(s,a,g))]$ for a convex function $f$. Throughout this paper, we choose $f(x) = \frac{1}{2}(x-1)^2$ as in Ma et al. [2022c], where the $f$-divergence is known as $\chi^2$-divergence under this specific choice of $f$ and it is shown to be more stable than other divergences such as KL divergence [Ma et al., 2022c]. Let $d_\alpha^*$ denote the optimal solution of (5), then the regularized optimal policy can be induced by $\pi_\alpha^* = \pi_{d_\alpha^*}$ as in (2). The following single-policy concentrability assumption assumes that $\pi_\alpha^*$ is well covered by the offline dataset.

**Assumption 1** (Single-policy concentrability for $\pi_\alpha^*$). *Let $d_\alpha^*$ be the optimal solution of (5), and let $\pi_\alpha^* = \pi_{d_\alpha^*}$ as defined in (2). We assume $C^{\pi_\alpha^*} \leq C_\alpha^*$ where $C^{\pi_\alpha^*}$ is defined in Definition 1 and $C_\alpha^* > 0$ is a constant.*

Under Assumption 1, it can be observed that the performance difference between the regularized optimal policy $\pi_\alpha^*$ and the optimal policy $\pi^*$ is bounded by $O(\alpha)$. The following proposition formally presents this observation.

**Proposition 3.1.** *Let $d_\alpha^*$ be the optimal solution of (5), and let $\pi_\alpha^* = \pi_{d_\alpha^*}$ as defined in (2). Then under Assumption 1, it holds that $J(\pi^*) - J(\pi_\alpha^*) \leq O\left(\alpha(C_\alpha^*)^2\right)$.*

The proof of Proposition 3.1 is deferred to Appendix A.1. Proposition 3.1 shows that by solving the regularized program (5), we can obtain a near-optimal policy as long as $\alpha$ is small. The algorithm of Ma et al. [2022c] also aims to solve (5) and they simply choose $\alpha = 1$. We show empirically in Section 5 that $\alpha < 1$ achieves better performance than $\alpha = 1$. In theory, we must carefully choose the value of $\alpha$ s.t. the suboptimality of our learned policy vanishes to 0 with a reasonable rate. Finally, as in Ma et al. [2022c], we convert (5) to the dual form, which is an unconstrained problem and amenable to solve:

**Proposition 3.2** (Dual form of (5)). *The duality form of (5) is*

$$\min_{V(s;g)\geq 0}(1-\gamma)\mathbb{E}_{(s,g)\sim(\rho,p(g))}[V(s;g)] + \mathbb{E}_{(s,a,g)\sim\mu}[\mathbb{1}\{g'_*(A_V(s,a;g)) \geq 0\}\bar{g}_*(A_V(s,a;g))]$$

$$\tag{6}$$

*where $g_*$ is the convex conjugate of $g = \alpha \cdot f$. Moreover, let $V_\alpha^*$ denote the optimal solution of (6), then it holds*

$$d_\alpha^*(s, a; g) = \mu(s, a; g)g_*'(r(s; g) + \gamma \mathcal{T} V_\alpha^*(s, a; g) - V_\alpha^*(s; g))_+ \tag{7}$$

*for all $(s, a, g) \in \mathcal{S} \times \mathcal{A} \times \mathcal{G}$.*

The proof of Proposition 3.2 is shown in Appendix A.2. According to the above proposition, one can first learn the $V$ function according to (6), and then use the learned $V$ function to learn the desired policy by (7). We call the first step *V-learning* and the second step *policy learning*, which will be discussed in detail in Sections 3.1 and 3.2 respectively. Finally, the main algorithm, which we call *VP-learning*, is presented in Algorithm 1.

---

**Algorithm 1** VP-learning

---

1: **Input:** Dataset $\mathcal{D} = \{(s_i, a_i, r_i, s_i', g_i)\}_{i=1}^N, \mathcal{D}_0 = \{(s_{0,i}, g_{0,i})\}_{i=1}^{N_0}$, value function class $\mathcal{V}$, policy class $\Pi$, model class $\mathcal{P}$ for stochastic settings.
2: Obtain $\hat{U}$ by *V-Learning* (Algorithm 2 or 3 ).
3: Obtain $\hat{\pi}$ by *policy learning* (Algorithm 4) using learned function $\hat{U}$.
4: **Output:** $\hat{\pi}$.

---

## 3.1 $V$-Learning

Define

$$L_\alpha(V) = \alpha((1 - \gamma)\mathbb{E}_{(s,g)\sim(\rho,p(g))}[V(s; g)] + \mathbb{E}_{(s,a,g)\sim\mu}[\mathbb{1}\{g_*'(A_V(s, a; g)) \geq 0\}\bar{g}_*(A_V(s, a; g))]). \tag{8}$$

Then (6) is equivalent to $\min_{V(s;g)\geq 0} L_\alpha(V)$. A natural estimator of $L_\alpha(V)$ is

$$\frac{1 - \gamma}{N_0} \sum_{i=1}^{N_0} \alpha \cdot V(s_{0,i}; g_{0,i}) + \frac{1}{N} \sum_{i=1}^N \alpha \cdot g_{*+}(r_i + \gamma V(s_i'; g_i) - V(s_i; g_i)). \tag{9}$$

However, when the transition kernel is not deterministic, this estimator is biased and will cause an over-estimation issue since $g_*(x) = \alpha f_*(x/\alpha) = \frac{\alpha(x/\alpha+1)^2}{2} - \frac{\alpha}{2}$ contains a square operator outside of the Bellman operator (consider estimating $(\mathbb{E}[X])^2$ using $\frac{1}{N}\sum X_i^2$).

Therefore, we use the original version of Ma et al. [2022c] for $V$-learning in deterministic dynamics (Algorithm 2 in Section 3.1.1), and a slightly modified version in stochastic dynamics (Algorithm 3 in Section 3.1.2). For both settings, we assume realizability of $V_\alpha^*$ on value function class $\mathcal{V}$:

**Assumption 2** (Realizability of $V_\alpha^*$). *Assume $V_\alpha^* \in \mathcal{V}$.*

### 3.1.1 $V$-Learning in Deterministic Dynamics

When the transition kernel $P$ is deterministic, it holds that $\mathcal{T}V(s, a; g) = V(s'; g)$ where $P(s'|s, a) = 1$. In this case, the natural estimator (9) is unbiased and can be directly applied to the $V$-learning procedure. The $V$-learning algorithm for deterministic dynamic settings is presented in Algorithm 2.

---

**Algorithm 2** $V$-learning in deterministic dynamics

---

1: **Input:** Dataset $\mathcal{D} = \{(s_i, a_i, r_i, s_i', g_i)\}_{i=1}^N, \mathcal{D}_0 = \{(s_{0,i}, g_{0,i})\}_{i=1}^{N_0}$, value function class $\mathcal{V}$.
2: $V$-learning by solving $\hat{V} = \arg\min_{V \in \mathcal{V}} \hat{L}^{(d)}(V)$ where

$$\hat{L}^{(d)}(V) \triangleq \frac{1 - \gamma}{N_0} \sum_{i=1}^{N_0} \alpha \cdot V(s_{0,i}; g_{0,i}) + \frac{\alpha}{N} \sum_{i=1}^N g_{*+}(r_i + \gamma V(s_i'; g_i) - V(s_i; g_i)). \tag{10}$$

3: $\hat{U}(s, a; g) \leftarrow r(s; g) + \gamma\hat{V}(s'; g) - \hat{V}(s; g) + \alpha$
4: **Output:** $\hat{V}, \hat{U}$.

---

Now for any $V$, we define $U_V(s, a; g) = r(s; g) + \gamma \mathcal{T}V(s, a; g) - V(s; g) + \alpha = A_V(s, a; g) + \alpha$ which can be interpreted as the advantage function of $V$ with an $\alpha$-shift. We also denote $U_\alpha^* = U_{V_\alpha^*}$.

Note that besides the learned $\hat{V}$ function, Algorithm 2 also outputs a $\hat{U}$ function. By (7), one can observe that in policy learning, what we indeed need is $\hat{U}$ instead of $\hat{V}$, and thus in the $V$-learning procedure we also compute this $\hat{U}$ function in preparation for policy learning.

One may challenge that $\hat{U}$ cannot be computed for all $(s, a; g)$ since we do not have knowledge of all $r(s; g)$. However, we only need the value of $\hat{U}(s_i, a_i; g_i)$ for $(s_i, a_i; g_i)$ contained in the offline dataset, where $r_i$ is also contained. Therefore, we can evaluate the value of $\hat{U}$ at all $(s, a; g)$ tuples requested in the policy learning algorithm.

Note that Algorithm 2 is equivalent to the first step of Ma et al. [2022c] except for the choice of $\alpha$ and a clip for the value of $g_*$. However, the above $V$-learning, as well as the original GoFAR algorithm, might suffer the over-estimation issue under stochastic dynamics, and we present algorithms suitable for stochastic dynamics in Section 3.1.2.

### 3.1.2 $V$-Learning in Stochastic Dynamics

When the transition kernel is stochastic, one cannot directly use $V(s'; g)$ to estimate $\mathcal{T}V(s, a; g)$. Since $\mathcal{T}V(s, a; g) = \mathbb{E}_{s' \sim P(\cdot|s,a)}[V(s'; g)]$, a natural idea is to learn the ground-truth transition kernel $P^{\star 1}$ first, and then use the learned transition kernel $\hat{P}$ to estimate $\mathcal{T}V(s, a; g)$: $\hat{\mathcal{T}}V(s, a; g) = \mathbb{E}_{s' \sim \hat{P}(\cdot|s,a)}[V(s'; g)]$. This is achievable under the following realizability assumption.

**Assumption 3** (Realizability of the ground-truth transition model). *Assume the ground-truth transition kernel $P^\star \in \mathcal{P}$.*

The algorithm for stochastic dynamic settings is presented in Algorithm 3, where we first learn the transition kernel $\hat{P}$, and then plug in the learned transition kernel to learn $V$ function. Similar to Algorithm 2, we also compute $\hat{U}$ in $V$-learning procedure.

---

**Algorithm 3** $V$-learning in stochastic dynamics

---

1: **Input:** Dataset $\mathcal{D} = \{(s_i, a_i, r_i, s'_i, g_i)\}_{i=1}^N, \mathcal{D}_0 = \{(s_{0,i}, g_{0,i})\}_{i=1}^{N_0}$, value function class $\mathcal{V}$, model class $\mathcal{P}$.
2: Estimate the transition kernel via maximum likelihood estimation (MLE)

$$\hat{P} = \max_{P \in \mathcal{P}} \frac{1}{N} \sum_{i=1}^{N} \log P(s'_i | s_i, a_i) \tag{11}$$

3: $V$-learning using the learned transition kernel: $\hat{V} = \arg\min_{V \in \mathcal{V}} \hat{L}^{(s)}(V)$ with

$$\hat{L}^{(s)}(V) \triangleq \frac{1-\gamma}{N_0} \sum_{i=1}^{N_0} \alpha \cdot V(s_{0,i}; g_{0,i}) + \frac{\alpha}{N} \sum_{i=1}^{N} g_{*+}(r_i + \gamma \hat{\mathcal{T}}V(s_i, a_i; g_i) - V(s_i; g_i)). \tag{12}$$

where $\hat{\mathcal{T}}V(s, a, ; g) = \mathbb{E}_{s' \sim \hat{P}(\cdot|s,a)}[V(s'; g)]$.
4: $\hat{U}(s, a; g) \leftarrow r(s; g) + \gamma \hat{\mathcal{T}}\hat{V}(s, a; g) - \hat{V}(s; g) + \alpha$
5: **Output:** $\hat{V}, \hat{U}$.

---

### 3.2 Policy Learning

We now derive policy learning, the second step of the VP-learning algorithm. Note that $\pi_\alpha^* = \arg\max_\pi \mathbb{E}_{(s,a,g) \sim d_\alpha^*}[\log \pi(a|s, g)]$. By (7), we also have

$$d_\alpha^*(s, a; g) = \mu(s, a; g) g'_*(r(s; g) + \gamma \mathcal{T}V_\alpha^*(s, a; g) - V_\alpha^*(s; g))_+ = \mu(s, a; g) \frac{U_\alpha^*(s, a; g)_+}{\alpha}.$$

---

[1]For notation convenience, in stochastic settings, we use $P^\star$ to denote the ground-truth transition kernel.

Therefore, $\pi_\alpha^* = \arg\max_\pi L_\alpha^{\mathrm{MLE}}(\pi)$ where $L_\alpha^{\mathrm{MLE}}(\pi) \triangleq \mathbb{E}_{(s,a,g)\sim\mu}\left[\frac{U_\alpha^*(s,a;g)_+}{\alpha}\log\pi(a|s,g)\right]$. Since we already learned $\hat{U}$, which is close to $U_\alpha^*$, we can use the following estimator for $L_\alpha^{\mathrm{MLE}}(\pi)$:

$$\hat{L}^{\mathrm{MLE}}(\pi) = \frac{1}{N}\sum_{i=1}^N \frac{\hat{U}(s_i,a_i;g_i)_+}{\alpha}\log\pi(a_i|s_i,g_i). \tag{13}$$

---

**Algorithm 4** Policy learning

1: **Input:** Dataset $\mathcal{D} = \{(s_i,a_i,r_i,s_i',g_i)\}_{i=1}^N$, policy class $\Pi$, $\hat{U}$ learned by Algorithm 2 or 3.
2: Policy learning by:

$$\hat{\pi} = \arg\max_{\pi\in\Pi}\hat{L}^{\mathrm{MLE}}(\pi) \triangleq \frac{1}{N}\sum_{i=1}^N \frac{\hat{U}(s_i,a_i;g_i)_+}{\alpha}\log\pi(a_i|s_i,g_i). \tag{14}$$

3: **Output:** $\hat{\pi}$.

---

The policy learning algorithm is presented in Algorithm 4, which can be viewed as a weighted maximum likelihood estimation (MLE) procedure. Finally, we make the following two assumptions on the policy class $\Pi$.

**Assumption 4** (Single-policy realizability). *Assume $\pi_\alpha^* \in \Pi$.*

**Assumption 5** (Lower bound of policy). *For any policy $\pi \in \Pi$, we assume that $\pi(a|s,g) \geq \tau > 0$ for any $(s,a,g) \in \mathcal{S}\times\mathcal{A}\times\mathcal{G}$.*

**Remark 1.** *One may consider Assumption 5 strong if $\tau$ is a constant independent of $\alpha$ or $N$. However, we allow that $\tau$ depends on $\alpha$. In that case, $\tau$ can be extremely small, and any policy mixed with a uniform policy with a tiny probability satisfies this assumption. Therefore, Assumption 5 is mild.*

## 4 Theoretical Guarantees

In this section, we provide theoretical guarantees of our main algorithm (Algorithm 1). We first show the results for $V$-Learning and policy learning in Section 4.1 and Section 4.2 respectively and then combine them to obtain our main theorem in Section 4.3.

### 4.1 Analysis of $V$-Learning

We mainly focus on $V$-learning in deterministic dynamics in this section. The analysis for stochastic dynamics is similar and presented in Appendix B.2.

As discussed in Section 3.1.1, although the first step of the algorithm is called $V$-learning, the main goal of this step is to estimate $U_\alpha^* = U_{V_\alpha^*}$ accurately. The following lemma provides a theoretical guarantee that the output of $V$-learning algorithm $\hat{U}$ is a good estimator of $U_\alpha^*$ in positive parts:

**Lemma 1** (Closeness of $\hat{U}_+$ and $U_{\alpha+}^*$). *Under Assumptions 1 and 2, with probability at least $1-\delta$, $\|\hat{U}_+ - U_{\alpha+}^*\|_{2,\mu} \leq O\left(\sqrt{\epsilon_{stat}}\right)$, where $\hat{U}$ is the output of Algorithm 2 and $\epsilon_{stat} \asymp V_{\max}^2\sqrt{\frac{\log(|\mathcal{V}|/\delta)}{N}}$.*

*Proof sketch.* By standard concentration argument, it can be shown that the empirical estimator $\hat{L}^{(d)}$ in Algorithm 2 (which is unbiased in deterministic dynamics) concentrates well on $L_\alpha$ for all $V \in \mathcal{V}$ (Lemma 3). Therefore, by realizability of $V_\alpha^*$, the value of $L_\alpha$ at $V_\alpha^*$ and the learned $V$-function $\hat{V}$ are close (Lemma 4). Finally, one can observe that $L_\alpha$ is "semi-strongly" convex w.r.t. $U_{V+}$ in $\|\cdot\|_{2,\mu}$-norm, and thus we can show that $\hat{U}_+$ and $U_{\alpha+}^*$ are also close. $\square$

The complete proof of Lemma 1 is deferred to Appendix B.1. In Appendix B.2, we also show the counterpart of Lemma 1 for stochastic dynamic settings.

## 4.2 Analysis of Policy Learning

After obtaining an accurate estimator $\hat{U}_+$ of $U^*_{\alpha+}$ in the $V$-Learning procedure, i.e., $\|\hat{U}_+ - U^*_{\alpha+}\|_{2,\mu} \lesssim \sqrt{\epsilon_{\text{stat}}}$, we can use $\hat{U}_+$ to perform policy learning and obtain the following guarantee:

**Lemma 2** (Closeness of $\pi^*_\alpha$ and $\hat{\pi}$). *Under Assumptions 4 and 5, with probability at least $1 - \delta$, the output policy $\hat{\pi}$ of Algorithm 4 satisfies*

$$\mathbb{E}_{s \sim d^*_\alpha, g \sim p(g)} \|\pi^*_\alpha(\cdot|s,g) - \hat{\pi}(\cdot|s,g)\|_{\text{TV}} \leq O\left(\sqrt{\epsilon^{MLE}_{stat}/\tau^2}\right),$$

*where $\epsilon^{MLE}_{stat}$ is defined in Lemma 8.*

The proof of Lemma 2 is provided in Appendix C.2. This result shows that the TV distance between the regularized optimal policy $\pi^*_\alpha$ and the output policy $\hat{\pi}$ by Algorithm 4 is small, which translates to a bounded performance difference between these two policies as formalized in Theorem 1.

**Theorem 1** (Suboptimality of $\hat{\pi}$). *Under Assumptions 4 and 5, with probability at least $1 - \delta$, the output policy $\hat{\pi}$ of Algorithm 4 satisfies $J(\pi^*_\alpha) - J(\hat{\pi}) \leq O\left(V_{\max}\sqrt{\epsilon^{MLE}_{stat}/\tau^2}\right)$.*

The proof of Theorem 1 is deferred to Appendix C.3.

## 4.3 Main Theorem: Statistical Rate of Suboptimality

Theorem 1 compares the performance difference between $\hat{\pi}$ and the regularized optimal policy $\pi^*_\alpha$. Since the ultimate goal is to compare with the optimal policy $\pi^*$, we also need to combine this result with Proposition 3.1. By carefully choosing the value of $\alpha$ to balance $J(\hat{\pi}) - J(\pi^*_\alpha)$ and $J(\pi) - J(\pi^*_\alpha)$, we can bound the suboptimality of the policy $\hat{\pi}$ output by Algorithm 1 compared to the optimal policy $\pi^*$, leading to the following main result:

**Theorem 2** (Statistical rate of suboptimality (in deterministic dynamics)). *Under Assumptions 1, 2, 4 and 5, with probability at least $1 - \delta$, the output policy $\hat{\pi}$ by Algorithm 1 (with the choice of Algorithm 2 for $V$-learning in deterministic dynamics) satisfies*

$$J(\pi^*) - J(\hat{\pi}) \lesssim \left(\frac{V^3_{\max}(C^*_\alpha)^3 \log(1/\tau)\log(|\mathcal{V}||\Pi|/\delta)}{\tau^2 N^{1/4}}\right)^{1/3}$$

*if we choose $\alpha \asymp \left(\frac{V^3_{\max}\log(1/\tau)\log(|\mathcal{V}||\Pi|/\delta)}{\tau^2(C^*_\alpha)^3 N^{1/4}}\right)^{1/3}$ and assume $N = N_0$.*

The proof of Theorem 2 is deferred to Appendix D.1. Note that Theorem 2 provides a suboptimality rate of $O(1/N^{1/12})$ which implies an $O(1/\text{poly}(\epsilon))$ sample complexity and thus is statistically efficient. A similar rate can also be obtained in stochastic dynamic settings, and we present the result in Appendix D.2. Note that our rate is slightly worse than the $O(1/N^{1/6})$ rate in Zhan et al. [2022], and worse than the optimal rate $O(1/\sqrt{N})$ in Rashidinejad et al. [2022]. We briefly discuss the intrinsic difficulty to derive an optimal convergence rate. First, we only require a realizability assumption on our function class, while Rashidinejad et al. [2022] requires a much stronger completeness assumption. Second, our optimization procedure is uninterleaved and only requires solving regression problems, while Zhan et al. [2022] and Rashidinejad et al. [2022] require solving minimax problems. Finally, Rashidinejad et al. [2022] assumes that the behavior policy is known and directly computes the policy using the knowledge of behavior policy, while our algorithm uses a more practical method, i.e., MLE, to solve the policy in the policy learning step.

We also compare our theoretical results to Ma et al. [2022c]. Theorem 4.1 of Ma et al. [2022c] provides a finite-sample guarantee for the suboptimality. However, they compare the performance of $\hat{\pi}$ to $\pi^*_\alpha$ (with $\alpha = 1$) instead of $\pi^*$. Since the performance gap between $\pi^*$ and $\pi^*_\alpha$ can be as large as a constant when $\alpha = 1$, even zero suboptimality (compared to $\pi^*_\alpha$) cannot imply that the learned policy has good performance. Moreover, their theoretical analysis assumes that $V^*$ can be learned with zero error, which is unreasonable in practical scenarios. We also note that they only provide a proof for deterministic policy classes, which can be restrictive in practice.

## 5 Experiments

In this section, we provide experimental results of our VP-learning algorithm with different choices of $\alpha$ under five different environments: FetchReach, FetchPick, FetchPush, FetchSlide, and HandReach [Plappert et al., 2018]. Similar to Ma et al. [2022c], the datasets for the five tasks are from Yang et al. [2022]. All the implementation details of our VP-learning are the same as GoFAR (see dataset details and implementation details in Ma et al. [2022c]) [2], except for the value of $\alpha$. Note that our VP-learning algorithm with $\alpha = 1$ is equivalent to the GoFAR algorithm. Table 1 presents the discounted returns and Table 2 presents the final distances of the policies trained after 100 epochs and evaluated over 10 runs. For each environment and each $\alpha$ , the result was averaged over 3 random seeds. The best results of each environment are in bold.

Table 1: Discounted return of different choices of $\alpha$, averaged over 3 random seeds.

| $\alpha \backslash$ Env | FetchReach | FetchPick | FetchPush | FetchSlide | HandReach |
|---|---|---|---|---|---|
| 0.01 | $27.4 \pm 0.29$ | $18.5 \pm 0.1$ | $18.0 \pm 1.8$ | $2.36 \pm 1.13$ | $8.72 \pm 1.69$ |
| 0.02 | $27.4 \pm 0.32$ | $18.7 \pm 1.8$ | $18.6 \pm 2.6$ | $2.40 \pm 0.47$ | $7.96 \pm 1.27$ |
| 0.05 | $27.4 \pm 0.32$ | $17.3 \pm 1.1$ | $19.3 \pm 2.0$ | $3.18 \pm 0.90$ | $\mathbf{8.98} \pm 3.11$ |
| 0.1 | $27.4 \pm 0.33$ | $20.3 \pm 1.3$ | $\mathbf{20.3} \pm 2.5$ | $3.22 \pm 0.38$ | $5.28 \pm 1.25$ |
| 0.2 | $27.4 \pm 0.32$ | $\mathbf{20.7} \pm 0.9$ | $17.7 \pm 2.9$ | $2.25 \pm 0.23$ | $2.92 \pm 0.98$ |
| 0.5 | $\mathbf{27.5} \pm 0.29$ | $18.5 \pm 0.4$ | $20.1 \pm 2.2$ | $\mathbf{3.47} \pm 1.08$ | $5.74 \pm 2.72$ |
| 1 | $27.3 \pm 0.34$ | $18.2 \pm 1.2$ | $19.6 \pm 1.6$ | $2.75 \pm 1.84$ | $7.13 \pm 3.60$ |
| 2 | $27.4 \pm 0.29$ | $18.3 \pm 0.7$ | $19.6 \pm 1.4$ | $1.80 \pm 0.66$ | $3.99 \pm 1.88$ |

Table 2: Final distance of different choices of $\alpha$, averaged over 3 random seeds.

| $\alpha \backslash$ Env | FetchReach | FetchPick | FetchPush | FetchSlide | HandReach |
|---|---|---|---|---|---|
| 0.01 | $0.0171 \pm 0.0017$ | $0.042 \pm 0.004$ | $0.033 \pm 0.001$ | $0.1177 \pm 0.012$ | $0.0269 \pm 0.0049$ |
| 0.02 | $0.0168 \pm 0.0016$ | $0.045 \pm 0.012$ | $0.031 \pm 0.002$ | $0.1085 \pm 0.010$ | $0.0274 \pm 0.0049$ |
| 0.05 | $0.0181 \pm 0.0011$ | $0.052 \pm 0.013$ | $0.032 \pm 0.002$ | $0.1061 \pm 0.009$ | $0.0270 \pm 0.0049$ |
| 0.1 | $0.0173 \pm 0.0014$ | $0.032 \pm 0.010$ | $\mathbf{0.027} \pm 0.002$ | $0.1018 \pm 0.002$ | $0.0275 \pm 0.0043$ |
| 0.2 | $0.0172 \pm 0.0019$ | $\mathbf{0.031} \pm 0.004$ | $0.031 \pm 0.003$ | $0.1029 \pm 0.010$ | $0.0275 \pm 0.0046$ |
| 0.5 | $\mathbf{0.0166} \pm 0.0011$ | $0.044 \pm 0.009$ | $0.031 \pm 0.005$ | $\mathbf{0.1017} \pm 0.017$ | $\mathbf{0.026826} \pm 0.0049$ |
| 1 | $0.0175 \pm 0.0013$ | $0.043 \pm 0.011$ | $0.043 \pm 0.012$ | $0.1202 \pm 0.019$ | $0.026828 \pm 0.0044$ |
| 2 | $0.0171 \pm 0.0011$ | $0.034 \pm 0.005$ | $0.032 \pm 0.001$ | $0.1044 \pm 0.011$ | $0.0275 \pm 0.0045$ |

The empirical results demonstrate the correctness of our theoretical analysis: choosing $\alpha = 1$ will result in a large suboptimality of $\pi_\alpha^*$ and thus the learned policy $\hat{\pi}$. Instead, we should carefully choose the value of $\alpha$ to ensure a vanishing suboptimality. In practice, we can tune the value of $\alpha$ and typically it is less than one. In our experiments, the best $\alpha$ ranges over $[0.05, 0.5]$.

## 6 Conclusions

In this paper, we theoretically analyze the VP-learning algorithm (Algorithm 1, which is based on the previous empirically successful algorithm in Ma et al. [2022c]) for both single-task and goal-conditioned offline settings. This algorithm can deal with general value function approximation and only requires near minimal assumptions on the dataset (single-policy concentrability) and function class (realizability). We also provide an $O(1/N^{1/12})$ upper bound of the suboptimality of the policy learned by the algorithm and empirically validate its effectiveness.

As for future directions, one important question is whether we can achieve the optimal suboptimality rate $\tilde{O}(1/\sqrt{N})$ while keeping the algorithm practical without unreasonably strong assumptions.

---

[2]We use the code at `https://github.com/JasonMa2016/GoFAR` with different values of $\alpha$ for our experiments.

## Acknowledgements

We thank the anonymous reviewer for catching a technical issue in a previous version of our paper. The work was done when HZ was a visiting researcher at Meta.

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

## A  Missing Proofs for Propositions in Section 3

### A.1  Proof of Proposition 3.1

*Proof.* Since both $d^*_\alpha$ and $d^*$ satisfy the Bellman flow constraint (1), due to the optimality of $d^*_\alpha$ in the regularized program (5), we have

$$\mathbb{E}_{(s,g)\sim d^*(s,g)}[r(s;g)] + \alpha D_f(d^*\|\mu) \le \mathbb{E}_{(s,g)\sim d^*_\alpha(s,g)}[r(s;g)] + \alpha D_f(d^*_\alpha\|\mu)$$

$$\implies \mathbb{E}_{(s,g)\sim d^*(s,g)}[r(s;g)] - \mathbb{E}_{(s,g)\sim d^*_\alpha(s,g)}[r(s;g)] \le \alpha D_f(d^*_\alpha\|\mu) \le \alpha(C^*_\alpha)^2/2.$$

Therefore,

$$J(\pi^*) - J(\pi^*_\alpha) = \mathbb{E}_{(s,g)\sim d^*(s,g)}[r(s;g)] - \mathbb{E}_{(s,g)\sim d^*_\alpha(s,g)}[r(s;g)] \le O\left(\alpha(C^*_\alpha)^2\right).$$

$\square$

### A.2  Proof of Proposition 3.2

*Proof.* By Lagrangian duality, we can obtain that the dual problem is

$$\min_{V(s;g)\ge 0} \max_{d(s,a;g)\ge 0} \mathbb{E}_{(s,g)\sim d(s,g)}[r(s;g)] - \alpha D_f(d(s,a;g)\|\mu(s,a;g))$$

$$+ \sum_{s,g} p(g)V(s;g)\left((1-\gamma)\rho(s) + \gamma \sum_{s',a'} P(s|s',a')d(s',a';g) - \sum_a d(s,a;g)\right) \tag{15}$$

where $p(g)V(s;g)$ is the Lagrangian vector. Similar to Ma et al. [2022c],

$$\sum_{s,g} p(g)V(s;g) \sum_{s',a'} P(s|s',a')d(s',a';g)$$

$$= \sum_{s',a',g} p(g)d(s',a';g) \sum_s P(s|s',a')V(s;g)$$

$$= \sum_{s',a',g} p(g)d(s',a';g)\mathcal{T}V(s',a';g).$$

Then (15) is equivalent to

$$\min_{V(s;g)\ge 0} \max_{d(s,a;g)\ge 0} (1-\gamma)\mathbb{E}_{(s,g)\sim(\rho,p(g))}[V(s;g)] + \mathbb{E}_{(s,a,g)\sim d}[r(s;g) + \gamma\mathcal{T}V(s,a;g) - V(s;g)]$$

$$-\alpha D_f(d(s,a;g)\|\mu(s,a;g)),$$

which can be further represented as

$$\min_{V(s;g)\ge 0} (1-\gamma)\mathbb{E}_{(s,g)\sim(\rho,p(g))}[V(s;g)] + \max_{d(s,a;g)\ge 0} \mathbb{E}_{(s,a,g)\sim d}[r(s;g) + \gamma\mathcal{T}V(s,a;g) - V(s;g)]$$

$$- D_g(d(s,a;g)\|\mu(s,a;g)),$$

where $g = \alpha \cdot f$. Combining the constraint that $d(s,a;g) \ge 0$ and the proof of Proposition 4.3 (Proposition B.2) of Ma et al. [2022c], the above program is equivalent to

$$\min_{V(s;g)\ge 0} (1-\gamma)\mathbb{E}_{(s,g)\sim(\rho,p(g))}[V(s;g)] + \mathbb{E}_{(s,a,g)\sim\mu}[\mathbb{1}\{g'_*(A_V(s,a;g)) \ge 0\}\bar{g}_*(A_V(s,a;g))],$$

and it holds that

$$d^*_\alpha(s,a;g) = \mu(s,a;g)g'_*(r(s;g) + \gamma\mathcal{T}V^*_\alpha(s,a;g) - V^*_\alpha(s;g))_+.$$

$\square$

## B  Proof for $V$-Learning

In this section, we provide theoretical analysis for the guarantees of $V$-learning in different settings (deterministic dynamics in Appendix B.1 and stochastic dynamics in Appendix B.2).

Recall that we choose $f(x) = \frac{(x-1)^2}{2}$, and thus $f_*(x) = \frac{(x+1)^2}{2} - \frac{1}{2}$. Therefore, we have $g(x) = \alpha f(x) = \frac{\alpha(x-1)^2}{2}$ and $g_*(x) = \frac{\alpha(x/\alpha+1)^2}{2} - \frac{\alpha}{2}$. Also, we define $g_{*,\max} = \max_{v \in [-V_{\max}, V_{\max}+1]} g_*(v)$, $g_{*,\min} = \min_{v \in [-V_{\max}, V_{\max}+1]} g_*(v)$ and $g_{*,\Delta} = g_{*,\max} - g_{*,\min}$. We can easily obtain that $g_{*,\Delta} \leq \alpha(1 + \frac{V_{\max}}{\alpha})^2 = O(V_{\max}^2/\alpha)$.

For the purpose of theoretical analysis, we further define

$$L_1(V) = (1-\gamma)\mathbb{E}_{(s,g)\sim(\rho,p(g))}[\alpha \cdot V(s;g)],$$
$$L_2(V) = \mathbb{E}_{(s,a,g)\sim\mu}[\alpha \cdot g_{*+}(r(s;g) + \gamma \mathcal{T}V(s,a;g) - V(s;g))],$$

and

$$\hat{L}_1^{(d)}(V) = \frac{1-\gamma}{N_0}\sum_{i=1}^{N_0} \alpha \cdot V(s_{0,i}; g_{0,i}),$$

$$\hat{L}_2^{(d)}(V) = \frac{1}{N}\sum_{i=1}^{N} \alpha \cdot g_{*+}(r(s_i; g_i) + \gamma V(s_i'; g_i) - V(s_i; g_i)).$$

Note that $L_\alpha(V) = L_1(V) + L_2(V)$ and $\hat{L}^{(d)}(V) = \hat{L}_1^{(d)}(V) + \hat{L}_2^{(d)}(V)$.

## B.1 Proof for Deterministic Dynamics

We first show that with high probability, the estimator $\hat{L}^{(d)}(V)$ in Algorithm 2 concentrates well on $L_\alpha(V)$ for all $V \in \mathcal{V}$.

**Lemma 3** (Concentration of $\hat{L}_1^{(d)}, \hat{L}_2^{(d)}$). *Under Assumptions 1 and 2, when the dynamic of the environment is deterministic, with probability at least $1 - \delta$, $\forall V \in \mathcal{V}$, it simultaneously holds that*

$$\left|\hat{L}_1^{(d)}(V) - L_1(V)\right| \leq O\left(\alpha V_{\max}\sqrt{\frac{\log(|\mathcal{V}|/\delta)}{N_0}}\right),$$

$$\left|\hat{L}_2^{(d)}(V) - L_2(V)\right| \leq O\left(\alpha \cdot g_{*,\Delta}\sqrt{\frac{\log(|\mathcal{V}|/\delta)}{N}}\right) = O\left(V_{\max}^2\sqrt{\frac{\log(|\mathcal{V}|/\delta)}{N}}\right),$$

*which immediately implies*

$$\left|\hat{L}^{(d)}(V) - L_\alpha(V)\right| \leq O\left(\alpha V_{\max}\sqrt{\frac{\log(|\mathcal{V}|/\delta)}{N_0}} + V_{\max}^2\sqrt{\frac{\log(|\mathcal{V}|/\delta)}{N}}\right) \triangleq \epsilon_{stat}.$$

*Proof.* First, fix any $V \in \mathcal{V}$. The expectation of $\hat{L}_1^{(d)}(V)$ is

$$\mathbb{E}[\hat{L}_1^{(d)}(V)] = \frac{1-\gamma}{N_0}\sum_{i=1}^{N_0} \mathbb{E}_{(s_{0,i},g_{0,i})\sim(\rho,p(g))}[\alpha \cdot V(s_{0,i}; g_{0,i})]$$
$$= (1-\gamma)\mathbb{E}_{(s,g)\sim(\rho,p(g))}[\alpha \cdot V(s;g)]$$
$$= L_1(V).$$

Also,

$$\mathbb{E}[\hat{L}_2^{(d)}(V)] = \frac{1}{N}\sum_{i=1}^{N} \mathbb{E}_{(s_i,a_i,g_i)\sim\mu}[\alpha \cdot g_{*+}(r(s_i; g_i) + \gamma V(s_i'; g_i) - V(s_i; g_i))]$$
$$= \frac{1}{N}\sum_{i=1}^{N} \mathbb{E}_{(s_i,a_i,g_i)\sim\mu}[\alpha \cdot g_{*+}(r(s_i; g_i) + \gamma \mathcal{T}V(s_i, a_i; g_i) - V(s_i; g_i))]$$
$$= L_2(V).$$

Also note that $V(s_{0,i}; g_{0,i}) \in [0, V_{\max}]$ and $g_{*+}(r(s_i; g_i) + \gamma \mathcal{T}V(s_i, a_i; g_i) - V(s_i; g_i)) \in [0, g_{*,\Delta}]$. By Hoeffding's inequality and a union bound, we have with probability at least $1 - \delta$,

$$\left| \hat{L}_1^{(d)}(V) - L_1(V) \right| \leq O\left( \alpha V_{\max} \sqrt{\frac{\log(1/\delta)}{N_0}} \right),$$

$$\left| \hat{L}_2^{(d)}(V) - L_2(V) \right| \leq O\left( \alpha \cdot g_{*,\Delta} \sqrt{\frac{\log(1/\delta)}{N}} \right).$$

Applying a union bound over all $V \in \mathcal{V}$ concludes the result. $\square$

Next, we show that the value of $L_\alpha$ at the regularized optimal $V$-function $V_\alpha^*$ and the learned function $\hat{V}$ is close.

**Lemma 4** (Closeness of the population objective of $\hat{V}$ and $V_\alpha^*$). *Under Assumptions 1 and 2, with probability at least $1 - \delta$, we have*

$$L_\alpha(\hat{V}) - L_\alpha(V_\alpha^*) \leq O(\epsilon_{stat}).$$

*Proof.* We condition on the high probability event in Lemma 3. Note that

$$L_\alpha(\hat{V}) - L_\alpha(V_\alpha^*) = \underbrace{L_\alpha(\hat{V}) - \hat{L}^{(d)}(\hat{V})}_{(1)} + \underbrace{\hat{L}^{(d)}(\hat{V}) - \hat{L}^{(d)}(V_\alpha^*)}_{(2)} + \underbrace{\hat{L}^{(d)}(V_\alpha^*) - L_\alpha(V_\alpha^*)}_{(3)}.$$

$(1), (3) \leq O(\epsilon_{stat})$ by Lemma 3 and $(2) \leq 0$ by the definition of $\hat{V}$, which completes the proof.

$\square$

Now we define "semi-strong" convexity, which is used to prove Lemma 1 and might be of independent interests.

**Definition 2** (Positive semi-strong convexity). *For an arbitrary set $\mathcal{X}$, assume $\mathcal{F} \subset \{f \mid f : \mathcal{X} \to \mathbb{R}\}$ is a convex set. Let $\rho \in \Delta(\mathcal{X})$ be a probability distribution over $\mathcal{X}$. The $\|\cdot\|_{2,\rho}$-norm of $f \in \mathcal{F}$ is defined as $\|f\|_{2,\rho} = \sqrt{\mathbb{E}_{x \sim \rho}[f^2(x)]}$. Define function $l : \mathbb{R} \to \mathbb{R}$ and define functional $\mathcal{L} : \mathcal{F} \to \mathbb{R}$ as*

$$\mathcal{L}(f) = \mathbb{E}_{x \sim \rho}[l(f(x))].$$

*We say $\mathcal{L}$ is $\sigma$-positive-semi-strongly convex w.r.t. $f$ in $\|\cdot\|_{2,\rho}$-norm for some $\sigma > 0$ if*

$$\mathcal{L}(f) - \frac{\sigma}{2}\|f_+\|_{2,\rho}^2$$

*is convex w.r.t. $f$ in $\|\cdot\|_{2,\rho}$-norm.*

Note that the common $\sigma$-strongly convex definition requires $\mathcal{L}(f) - \frac{\sigma}{2}\|f\|_{2,\rho}^2$ to be convex. Now we prove the following property for $\sigma$-positive-semi-strong convexity.

**Proposition B.1.** *Under the same setting of Definition 2, if $\mathcal{L}$ is $\sigma$-positive-semi-strongly convex w.r.t. $f$ in $\|\cdot\|_{2,\rho}$-norm, it holds that for any $f, g \in \mathcal{F}$,*

$$\mathcal{L}(f) - \mathcal{L}(g) \geq \nabla\mathcal{L}(g)^\mathsf{T}(f - g) + \frac{\sigma}{2}\|f_+ - g_+\|_{2,\rho}^2.$$

*Proof.* By Definition 2, $\tilde{\mathcal{L}}(f) \triangleq \mathcal{L}(f) - \frac{\sigma}{2}\|f_+\|_{2,\rho}^2$ is convex, which, by the definition of convex, implies that

$$\tilde{\mathcal{L}}(f) - \tilde{\mathcal{L}}(g) \geq \nabla\tilde{\mathcal{L}}(g)^\mathsf{T}(f - g).$$

Plugging in the definition of $\tilde{\mathcal{L}}(\cdot)$, we have

$$\mathcal{L}(f) - \frac{\sigma}{2}\|f_+\|_{2,\rho}^2 - \mathcal{L}(g) + \frac{\sigma}{2}\|g_+\|_{2,\rho}^2 \geq \nabla\mathcal{L}(g)^\mathsf{T}(f - g) - \sigma\mathbb{E}_{x \sim \rho}[g(x)_+(f(x) - g(x))].$$

Rearranging, we can obtain that

$$
\begin{aligned}
\mathcal{L}(f) - \mathcal{L}(g) \geq & \nabla\mathcal{L}(g)^\mathsf{T}(f-g) - \sigma\mathbb{E}_{x\sim\rho}[g(x)_+(f(x)-g(x))] + \frac{\sigma}{2}\|f_+\|_{2,\rho}^2 - \frac{\sigma}{2}\|g_+\|_{2,\rho}^2 \\
= & \nabla\mathcal{L}(g)^\mathsf{T}(f-g) + \frac{\sigma}{2}\mathbb{E}_{x\sim\rho}[f_+^2(x) - g_+^2(x) - 2g(x)_+(f(x)-g(x))] \\
= & \nabla\mathcal{L}(g)^\mathsf{T}(f-g) + \frac{\sigma}{2}\mathbb{E}_{x\sim\rho}[f_+^2(x) - g_+^2(x) - 2g(x)_+(f(x)-g(x))] \\
= & \nabla\mathcal{L}(g)^\mathsf{T}(f-g) + \frac{\sigma}{2}\|f_+ - g_+\|_{2,\rho}^2 \\
& + \sigma\mathbb{E}_{x\sim\rho}[g_+(x)(f_+(x)-f(x))] + \sigma\mathbb{E}_{x\sim\rho}[g_+(x)(g(x)-g_+(x))].
\end{aligned}
$$

It suffices to show that

$$
\mathbb{E}_{x\sim\rho}[g_+(x)(f_+(x)-f(x))] \geq 0, \quad \mathbb{E}_{x\sim\rho}[g_+(x)(g(x)-g_+(x))] \geq 0.
$$

Note that for any $x \in \mathcal{X}$, $g_+(x) \geq 0$ and $f_+ \geq f(x)$, which implies $g_+(x)(f_+(x) - f(x)) \geq 0$. Also, note that when $g(x) \geq 0$, we have $g(x) - g_+(x) = 0$, and when $g(x) < 0$, we have $g_+(x) = 0$, which means $g_+(x)(g(x) - g_+(x)) \equiv 0$. Therefore, we have

$$
\mathbb{E}_{x\sim\rho}[g_+(x)(f_+(x)-f(x))] \geq 0, \quad \mathbb{E}_{x\sim\rho}[g_+(x)(g(x)-g_+(x))] = 0,
$$

which concludes. $\qquad\square$

Finally, by semi-strong convexity of $L_\alpha$ w.r.t. $U_V$ in $\|\cdot\|_{2,\mu}$-norm, we can show that $\hat{U}$ and $U_\alpha^*$ are also close.

*Proof of Lemma 1.* We condition on the high probability event in Lemma 3. Recall that $U_V(s, a; g) = r(s; g) + \gamma\mathcal{T}V(s, a; g) - V(s; g) + \alpha$ and in deterministic dynamics, $\hat{U} = U_{\hat{V}}$. Let $\mathcal{U} = \{U_V : V \in \mathbb{R}_+^{|\mathcal{S}|\times|\mathcal{G}|}\} \subseteq \mathbb{R}^{|\mathcal{S}|\times|\mathcal{A}|\times|\mathcal{G}|}$ which is a convex set by definition. Also, since $U_V$ is linear in $V$, we can also obtain that $V$ can be linearly represented by $U$. Therefore, we can define that $\tilde{L}_\alpha(U_V) = L_\alpha(V)$ and $\tilde{L}_\alpha(U_V) - \frac{1}{2}\mathbb{E}_{(s,a,g)\sim\mu}[U_V^2(s, a, g)_+]$ is linear and thus convex in $U_V$, which implies that $\tilde{L}_\alpha(U_V)$ is 1-positive-semi-strongly convex w.r.t. $U_V$ and $\|\cdot\|_{2,\mu}$. Then, by Proposition B.1, we have

$$
\tilde{L}_\alpha(\hat{U}) - \tilde{L}_\alpha(U_\alpha^*) \geq \nabla\tilde{L}_\alpha(U_\alpha^*)^\mathsf{T}(\hat{U} - U_\alpha^*) + \frac{1}{2}\|\hat{U}_+ - U_{\alpha+}^*\|_{2,\mu}^2.
$$

Since $U_\alpha^* = \arg\min_{U\in\mathcal{U}} \tilde{L}_\alpha(U)$, by the first order optimality condition, it holds that $\nabla\tilde{L}_\alpha(U_\alpha^*)^\mathsf{T}(\hat{U} - U_\alpha^*) \geq 0$. Combining Lemma 4, we have

$$
\|\hat{U}_+ - U_{\alpha+}^*\|_{2,\mu} \leq \sqrt{2(\tilde{L}_\alpha(\hat{U}) - \tilde{L}_\alpha(U_\alpha^*))} \leq O\left(\sqrt{\epsilon_{\text{stat}}}\right).
$$

$\qquad\square$

## B.2 Proof for Stochastic Dynamics

In stochastic dynamic settings, we first learn the ground-truth transition model and then calculate $\hat{U}$. The following lemma provides a theoretical guarantee for maximum likelihood estimation (MLE) that $\hat{P}$ and $P^\star$ are close.

**Lemma 5** (Convergence rate of MLE, Van de Geer [2000]). *For any fixed $\delta > 0$, with probability at least $1 - \delta$, we have*

$$
\mathbb{E}_{(s,a,g)\sim\mu}\left[\|\hat{P}(\cdot|s, a) - P^\star(\cdot|s, a)\|_{\mathsf{TV}}^2\right] \lesssim \frac{\log(|\mathcal{P}|/\delta)}{N},
$$

*where $\hat{P}$ is defined as in* (11). *This immediately implies that*

$$
\mathbb{E}_{(s,a,g)\sim\mu}\left[\|\hat{P}(\cdot|s, a) - P^\star(\cdot|s, a)\|_{\mathsf{TV}}\right] \lesssim \sqrt{\frac{\log(|\mathcal{P}|/\delta)}{N}}.
$$

Equipped with Lemma 5, we can guarantee that for any $V$, the population objective $L_\alpha(V)$ is close to the empirical objective $\hat{L}^{(s)}(V)$, which is presented in Lemma 6.

**Lemma 6** (Concentration of $\hat{L}^{(s)}$). *Under Assumptions 1 to 3, with probability at least $1-\delta$, $\forall V \in \mathcal{V}$, it holds that*

$$\left| \hat{L}^{(s)}(V) - L_\alpha(V) \right|$$

$$\leq O\left( \alpha V_{\max} \sqrt{\frac{\log(|\mathcal{V}|/\delta)}{N_0}} + V_{\max}^2 \sqrt{\frac{\log(|\mathcal{V}|/\delta)}{N}} + \gamma V_{\max}^2 \sqrt{\frac{\log(|\mathcal{P}|/\delta)}{N}} \right) \triangleq \epsilon_{stat}^{stochastic}.$$

*Proof.* For convenience, define

$$\hat{L}_1^{(s)}(V) = \frac{1-\gamma}{N_0} \sum_{i=1}^{N_0} \alpha \cdot V(s_{0,i}; g_{0,i}),$$

$$\hat{L}_2^{(s)}(V) = \frac{1}{N} \sum_{i=1}^{N} \alpha \cdot g_{*+}(r(s_i; g_i) + \gamma \hat{\mathcal{T}} V(s_i, a_i; g_i) - V(s_i; g_i)).$$

Note that $\hat{L}_1^{(s)}(V) = \hat{L}_1^{(d)}(V)$ and thus by Lemma 3 we have that with probability at least $1-\delta$, for all $V \in \mathcal{V}$,

$$\left| \hat{L}_1^{(s)}(V) - L_1(V) \right| \leq O\left( \alpha V_{\max} \sqrt{\frac{\log(|\mathcal{V}|/\delta)}{N_0}} \right). \tag{16}$$

Now define

$$\tilde{L}_2^{(s)}(V) = \mathbb{E}_{(s,a,g)\sim\mu}[\alpha \cdot g_{*+}(r(s; g) + \gamma \hat{\mathcal{T}} V(s, a; g) - V(s; g))]$$

Since $\mathbb{E}_{(s,a,g)\sim\mu}[\hat{L}_2^{(s)}(V)] = \tilde{L}_2^{(s)}(V)$, by Lemma 3, we have that with probability $1-\delta$, for any $V \in \mathcal{V}$, it holds that

$$|\hat{L}_2^{(s)}(V) - \tilde{L}_2^{(s)}(V)| \leq O\left( V_{\max}^2 \sqrt{\frac{\log(|\mathcal{V}|/\delta)}{N}} \right). \tag{17}$$

Also, note that

$$|L_2(V) - \tilde{L}_2^{(s)}(V)|$$

$$= \left| \alpha \mathbb{E}_{(s,a,g)\sim\mu}[g_{*+}(r(s; g) + \gamma \mathcal{T} V(s, a; g) - V(s; g)) - g_{*+}(r(s; g) + \gamma \hat{\mathcal{T}} V(s, a; g) - V(s; g))] \right|$$

$$= \frac{1}{2} \left| \mathbb{E}_{(s,a,g)\sim\mu} \left[ (r(s; g) + \gamma \mathcal{T} V(s, a; g) - V(s; g) + \alpha)_+^2 - (r(s; g) + \gamma \hat{\mathcal{T}} V(s, a; g) - V(s; g) + \alpha)_+^2 \right] \right|$$

$$\lesssim V_{\max} \mathbb{E}_{(s,a,g)\sim\mu}[\gamma \cdot |(\mathcal{T} - \hat{\mathcal{T}}) V(s, a; g)|]$$

$$= V_{\max} \mathbb{E}_{(s,a,g)\sim\mu}[\gamma \cdot |\mathbb{E}_{s'\sim P^\star(\cdot|s,a)}[V(s'; g)] - \mathbb{E}_{s'\sim\hat{P}(\cdot|s,a)}[V(s'; g)]|]$$

$$\lesssim \gamma V_{\max}^2 \mathbb{E}_{(s,a,g)\sim\mu}[\|P^\star(\cdot|s, a) - \hat{P}(\cdot|s, a)\|_{\mathsf{TV}}].$$

By Lemma 5, with probability at least $1-\delta$, for all $V \in \mathcal{V}$,

$$|L_2(V) - \tilde{L}_2^{(s)}(V)| \lesssim \gamma V_{\max}^2 \mathbb{E}_{(s,a,g)\sim\mu}[\|P^\star(\cdot|s, a) - \hat{P}(\cdot|s, a)\|_{\mathsf{TV}}] \lesssim \gamma V_{\max}^2 \sqrt{\frac{\log(|\mathcal{P}|/\delta)}{N}}. \tag{18}$$

By rescaling $\delta$ and applying a union bound, we can obtain that with probability at least $1-\delta$, (16), (17), (18) hold simultaneously. The conclusion holds by applying the triangle inequality. $\square$

Finally, we show that $\hat{U}_+$ and $U^*_{\alpha+}$ are close.

**Lemma 7** (Closeness of $\hat{U}_+$ and $U_{\alpha+}^*$ in stochastic dynamics). *Under Assumptions 1 to 3, with probability at least $1 - \delta$,*

$$\|\hat{U}_+ - U_{\alpha+}^*\|_{2,\mu} \leq O\left(\sqrt{\epsilon_{stat}^{stochastic}}\right).$$

*where $\hat{U}$ is the output of Algorithm 3, $\alpha = \Omega\left(\frac{\log(|\mathcal{V}||\mathcal{P}|/\delta)}{\sqrt{N}}\right)$ and*

$$\epsilon_{stat}^{stochastic} \asymp V_{\max}^2 \sqrt{\frac{\log(|\mathcal{V}||\mathcal{P}|/\delta)}{N}}.$$

*Proof.* For convenience, let $\tilde{U} = U_{\hat{V}}$. Following the same analysis as in the deterministic case, we have $\|U_{\alpha+}^* - \tilde{U}_+\|_{2,\mu} \leq O\left(\sqrt{\epsilon_{stat}^{stochastic}}\right)$ by Lemma 6. Also,

$$
\begin{aligned}
\|\tilde{U}_+ - \hat{U}_+\|_{2,\mu}^2 &\leq \|\tilde{U} - \hat{U}\|_{2,\mu}^2 \\
&= \mathbb{E}_{(s,a,g)\sim\mu}\left[\left(\gamma\mathbb{E}_{s'\sim P^\star(\cdot|s,a)}[\hat{V}(s;g)] - \gamma\mathbb{E}_{s'\sim\hat{P}(\cdot|s,a)}[\hat{V}(s;g)]\right)^2\right] \\
&\lesssim \gamma^2 V_{\max}^2 \mathbb{E}_{(s,a,g)\sim\mu}\left[\|\hat{P}(\cdot|s,a) - P^\star(\cdot|s,a)\|_{\mathsf{TV}}^2\right] \\
&\lesssim \gamma^2 V_{\max}^2 \frac{\log(|\mathcal{P}|/\delta)}{N},
\end{aligned}
$$

where the last inequality holds by Lemma 5. Therefore,

$$
\begin{aligned}
\|\hat{U}_+ - U_{\alpha+}^*\|_{2,\mu} &\leq \|\tilde{U}_+ - U_{\alpha+}^*\|_{2,\mu} + \|\tilde{U}_+ - \hat{U}_+\|_{2,\mu} \\
&\lesssim \sqrt{\epsilon_{stat}^{stochastic}} + \gamma V_{\max}\sqrt{\frac{\log(|\mathcal{P}|/\delta)}{N}} \\
&\lesssim \sqrt{\epsilon_{stat}^{stochastic}}.
\end{aligned}
$$

$\square$

## C Proof for Policy Learning

We provide a theoretical analysis of policy learning in this section. We first show two key lemmas in Appendix C.1, and then provide missing proofs of the main text in Appendix C.2 and Appendix C.3.

### C.1 Key Lemmas

**Lemma 8** (Statistical error of the weighted MLE objective). *Under Assumption 5, with probability at least $1 - \delta$, for any policy $\pi \in \Pi$, it holds that*

$$|L_\alpha^{MLE}(\pi) - \hat{L}^{MLE}(\pi)| \leq (\epsilon_U + \epsilon_\Pi)\log(1/\tau) \triangleq \epsilon_{stat}^{MLE}.$$

*where $\epsilon_U \triangleq O(\sqrt{\epsilon_{stat}/\alpha^2})$ and $\epsilon_\Pi \triangleq O\left(C_\alpha^*\sqrt{\frac{\log(|\Pi|/\delta)}{N}} + \frac{V_{\max}}{\alpha N}\log(|\Pi|/\delta)\right)$.*

*Proof.* For convenience, we assume that the offline dataset used in policy learning is independent of the dataset used in $V$-Learning. This can be easily achieved by splitting the original dataset $\mathcal{D}$ uniformly at random into two datasets of equal size. Also, for analysis, we define

$$\tilde{L}^{\mathrm{MLE}}(\pi) = \mathbb{E}_{(s,a,g)\sim\mu}\left[\frac{\hat{U}(s,a;g)_+}{\alpha}\log\pi(a|s,g)\right], \quad \forall \pi \in \Pi.$$

Now we condition on the high probability event in Lemma 3, which we denote by $\mathcal{E}_1$. Then by Lemma 1, we have $\left\|\frac{\hat{U}_+}{\alpha} - \frac{U_{\alpha+}^*}{\alpha}\right\|_{2,\mu} \leq \epsilon_U$. By Cauchy-Schwarz inequality, we can obtain that for

any policy $\pi$,

$$|L_\alpha^{\text{MLE}}(\pi) - \tilde{L}^{\text{MLE}}(\pi)| \leq \mathbb{E}_{(s,a,g)\sim\mu}\left[\left|\frac{U_\alpha^*(s,a;g)_+ - \hat{U}(s,a;g)_+}{\alpha}\right| |\log\pi(a|s,g)|\right]$$

$$\leq \left\|\frac{\hat{U}_+}{\alpha} - \frac{U_{\alpha+}^*}{\alpha}\right\|_{2,\mu} \|\log\pi(a|s,g)\|_{2,\mu}$$

$$\leq \epsilon_U \log(1/\tau).$$

Also, for any fixed $\pi \in \Pi$, since $\mathbb{E}_{(s,a,g)\sim\mu}[\hat{L}^{\text{MLE}}(\pi)] = \tilde{L}^{\text{MLE}}(\pi)$, we can obtain by Bernstein's inequality that with probability at least $1 - \delta$,

$$|\tilde{L}^{\text{MLE}}(\pi) - \hat{L}^{\text{MLE}}(\pi)|$$

$$\leq O\left(\sqrt{\frac{\text{Var}_\mu\left(\frac{\hat{U}(s,a;g)_+}{\alpha}\log\pi(a|s,g)\right)\log(1/\delta)}{N}} + \frac{\|\hat{U}_+/\alpha\|_\infty\log(1/\tau)}{N}\log(1/\delta)\right).$$

Since $U_\alpha^*(s,a;g)_+/\alpha = d_\alpha^*(s,a;g)/\mu(s,a;g) \leq C_\alpha^*$, we have $\|U_{\alpha+}^*/\alpha\|_{2,\mu} \leq C_\alpha^*$ and thus $\|\hat{U}_+/\alpha\|_{2,\mu} \leq \|U_{\alpha+}^*/\alpha\|_{2,\mu} + \epsilon_U \leq C_\alpha^* + \epsilon_U \leq O(C_\alpha^*)$. Also,

$$\text{Var}_\mu\left(\frac{\hat{U}(s,a;g)_+}{\alpha}\log\pi(a|s,g)\right) \leq \mathbb{E}_\mu\left[\left(\frac{\hat{U}(s,a;g)_+}{\alpha}\right)^2\log^2\pi(a|s,g)\right]$$

$$\leq \mathbb{E}_\mu\left[\left(\frac{\hat{U}(s,a;g)_+}{\alpha}\right)^2\right]\log^2(1/\tau)$$

$$\leq O((C_\alpha^*)^2\log^2(1/\tau)).$$

Applying a union bound over all $\pi \in \Pi$, it holds that with probability at least $1 - \delta$, for all $\pi \in \Pi$

$$|\tilde{L}^{\text{MLE}}(\pi) - \hat{L}^{\text{MLE}}(\pi)|$$

$$\leq O\left(C_\alpha^*\log(1/\tau)\sqrt{\frac{\log(|\Pi|/\delta)}{N}} + \frac{V_{\max}\log(1/\tau)}{\alpha N}\log(|\Pi|/\delta)\right)$$

$$= \epsilon_\Pi\log(1/\tau).$$

We denote the above event by $\mathcal{E}_2$. When $\mathcal{E}_1$ and $\mathcal{E}_2$ hold simultaneously, we have by the triangle inequality that

$$|L_\alpha^{\text{MLE}}(\pi) - \hat{L}^{\text{MLE}}(\pi)| \leq |L_\alpha^{\text{MLE}}(\pi) - \tilde{L}^{\text{MLE}}(\pi)| + |\tilde{L}^{\text{MLE}}(\pi) - \hat{L}^{\text{MLE}}(\pi)| \leq (\epsilon_U + \epsilon_\Pi)\log(1/\tau).$$

Also,

$$\mathbb{P}(\neg(\mathcal{E}_1 \cap \mathcal{E}_2)) = \mathbb{P}(\neg\mathcal{E}_1 \cup \neg\mathcal{E}_2) \leq \mathbb{P}(\neg\mathcal{E}_1) + \mathbb{P}(\neg\mathcal{E}_2)$$

$$\leq \delta + \mathbb{P}(\mathcal{E}_1)\mathbb{P}(\neg\mathcal{E}_2|\mathcal{E}_1) + \mathbb{P}(\neg\mathcal{E}_1)\mathbb{P}(\neg\mathcal{E}_2|\neg\mathcal{E}_1)$$

$$\leq \delta + 1 \times \delta + \delta \times 1 \leq 3\delta.$$

The conclusion holds by rescaling $\delta$. $\qquad\square$

**Lemma 9** (Closeness of MLE objective of $\pi_\alpha^*$ and $\hat{\pi}$)**.** *Under Assumption 4, with probability at least $1 - \delta$,*

$$L_\alpha^{MLE}(\pi_\alpha^*) - L_\alpha^{MLE}(\hat{\pi}) \leq O(\epsilon_{stat}^{MLE}).$$

*Proof.* We condition on the high probability event in Lemma 8. Note that

$$L_\alpha^{\text{MLE}}(\pi_\alpha^*) - L_\alpha^{\text{MLE}}(\hat{\pi}) = \underbrace{L_\alpha^{\text{MLE}}(\pi_\alpha^*) - \hat{L}^{\text{MLE}}(\pi_\alpha^*)}_{(1)} + \underbrace{\hat{L}^{\text{MLE}}(\pi_\alpha^*) - \hat{L}^{\text{MLE}}(\hat{\pi})}_{(2)} + \underbrace{\hat{L}^{\text{MLE}}(\hat{\pi}) - L_\alpha^{\text{MLE}}(\hat{\pi})}_{(3)}.$$

$(1), (3) \leq O(\epsilon_{\text{stat}}^{\text{MLE}})$ by Lemma 8 and $(2) \leq 0$ by the optimality of $\hat{\pi}$ in empirical MLE objective, which completes the proof. $\qquad\square$

## C.2 Proof of Lemma 2

*Proof of Lemma 2.* We condition on the high probability event in Lemma 8. Note that

$$L_\alpha^{\text{MLE}}(\pi) = \mathbb{E}_{(s,a,g)\sim\mu}[g'_*(r(s;g) + \gamma\mathcal{T}V_\alpha^*(s,a;g) - V_\alpha^*(s;g))_+ \log\pi(a|s,g)]$$
$$= \mathbb{E}_{(s,a,g)\sim d_\alpha^*}[\log\pi(a|s,g)].$$

We also define

$$L_{\alpha,\text{Rel}}^{\text{MLE}}(\pi) = L_\alpha^{\text{MLE}}(\pi) - \mathbb{E}_{(s,a,g)\sim d_\alpha^*}[\log\pi_\alpha^*(a|s,g)] = \mathbb{E}_{(s,a,g)\sim d_\alpha^*}\left[\log\frac{\pi(a|s,g)}{\pi_\alpha^*(a|s,g)}\right],$$

and note that $L_{\alpha,\text{Rel}}^{\text{MLE}}$ is a constant shift of $L_\alpha^{\text{MLE}}$. For any $\pi \in \Pi$, we further define $r_\pi = \pi/\pi_\alpha^*$. By Assumption 5, $r_\pi(a|s,g) \in [\tau, 1/\tau]$. Let $\tilde{L}_{\alpha,\text{Rel}}^{\text{MLE}}(r_\pi) = \mathbb{E}_{(s,a,g)\sim d_\alpha^*}[\log r_\pi(a|s,g)]$ which is $2\tau^2$-strongly concave w.r.t. $\|\cdot\|_{2,d_\alpha^*}$ when $r_\pi(a|s,g) \in [\tau, 1/\tau]$. Since $\pi_\alpha^*$ is the maximizer of $L_\alpha^{\text{MLE}}$ and thus the maximizer of $L_{\alpha,\text{Rel}}^{\text{MLE}}$, we have that $r_{\pi_\alpha^*}$ is the maximizer of $\tilde{L}_{\alpha,\text{Rel}}^{\text{MLE}}$. By strong concavity and the optimality of $r_{\pi_\alpha^*}$, we have

$$\tau^2\|r_{\pi_\alpha^*} - r_{\hat\pi}\|_{2,d_\alpha^*}^2 \le \tilde{L}_{\alpha,\text{Rel}}^{\text{MLE}}(r_{\pi_\alpha^*}) - \tilde{L}_{\alpha,\text{Rel}}^{\text{MLE}}(r_{\hat\pi})$$
$$= L_{\alpha,\text{Rel}}^{\text{MLE}}(\pi_\alpha^*) - L_{\alpha,\text{Rel}}^{\text{MLE}}(\hat\pi) = L_\alpha^{\text{MLE}}(\pi_\alpha^*) - L_\alpha^{\text{MLE}}(\hat\pi) \le O(\epsilon_{\text{stat}}^{\text{MLE}}),$$

where the last inequality holds by Lemma 9. Note that

$$\|r_{\pi_\alpha^*} - r_{\hat\pi}\|_{2,d_\alpha^*}^2 = \mathbb{E}_{(s,a,g)\sim d_\alpha^*}\left[\left(r_{\hat\pi}(a|s,g) - r_{\pi_\alpha^*}(a|s,g)\right)^2\right]$$
$$= \mathbb{E}_{(s,a,g)\sim d_\alpha^*}\left[\left(\frac{\hat\pi(a|s,g)}{\pi_\alpha^*(a|s,g)} - 1\right)^2\right]$$
$$\gtrsim \mathbb{E}_{(s,g)\sim d_\alpha^*}\left[\|\hat\pi(\cdot|s,g) - \pi_\alpha^*(\cdot|s,g)\|_{\text{TV}}^2\right]$$
$$\ge \left(\mathbb{E}_{(s,g)\sim d_\alpha^*}\left[\|\hat\pi(\cdot|s,g) - \pi_\alpha^*(\cdot|s,g)\|_{\text{TV}}\right]\right)^2$$

where the last inequality holds since TV distance is upper bounded by $\chi^2$ distance. Therefore, we can finally obtain that

$$\mathbb{E}_{(s,g)\sim d_\alpha^*}\left[\|\hat\pi(\cdot|s,g) - \pi_\alpha^*(\cdot|s,g)\|_{\text{TV}}\right] \le O\left(\sqrt{\epsilon_{\text{stat}}^{\text{MLE}}/\tau^2}\right).$$

$\square$

## C.3 Proof of Theorem 1

*Proof of Theorem 1.* We condition on the high probability event in Lemma 8. By performance difference lemma [Agarwal et al., 2019], we have

$$J(\pi_\alpha^*) - J(\hat\pi) = \mathbb{E}_{(s,a,g)\sim d_\alpha^*}[A^{\hat\pi}(s,a;g)]$$
$$= \mathbb{E}_{(s,g)\sim d_\alpha^*}[\mathbb{E}_{a\sim\pi_\alpha^*(\cdot|s,g)}A^{\hat\pi}(s,a;g) - \mathbb{E}_{a\sim\hat\pi(\cdot|s,g)}A^{\hat\pi}(s,a;g)]$$
$$\lesssim V_{\max}\mathbb{E}_{(s,g)\sim d_\alpha^*}[\|\pi_\alpha^*(\cdot|s,g) - \hat\pi(\cdot|s,g)\|_1]$$
$$\lesssim V_{\max}\mathbb{E}_{(s,g)\sim d_\alpha^*}[\|\pi_\alpha^*(\cdot|s,g) - \hat\pi(\cdot|s,g)\|_{\text{TV}}]$$
$$\lesssim V_{\max}\sqrt{\frac{\epsilon_{\text{stat}}^{\text{MLE}}}{\tau^2}},$$

where the last inequality holds by Lemma 2.

$\square$

# D   Statistical Rate of the Suboptimality in Different Settings

In this section, we analyze the statistical rate of the suboptimality of the output policy $\hat\pi$ by Algorithm 1 in different settings.

## D.1 Deterministic Settings

*Proof of Theorem 2.* By Theorem 1 and Proposition 3.1,

$$
\begin{aligned}
& J(\pi^*) - J(\hat{\pi}) \\
={} & J(\pi^*) - J(\pi_\alpha^*) + J(\pi_\alpha^*) - J(\hat{\pi}) \\
\lesssim{} & \alpha(C_\alpha^*)^2 + V_{\max}\sqrt{\frac{\epsilon_{\text{stat}}^{\text{MLE}}}{\tau^2}} \\
\lesssim{} & \alpha(C_\alpha^*)^2 + \frac{V_{\max}\sqrt{\log(1/\tau)}}{\tau}\sqrt{\sqrt{\epsilon_{\text{stat}}/\alpha^2} + C_\alpha^*\sqrt{\frac{\log(|\Pi|/\delta)}{N}} + \frac{V_{\max}}{\alpha N}\log(|\Pi|/\delta)}.
\end{aligned}
$$

Since $\epsilon_{\text{stat}} \asymp V_{\max}^2\sqrt{\frac{\log(|\mathcal{V}|/\delta)}{N}}$, we can further obtain that

$$
\begin{aligned}
& J(\pi^*) - J(\hat{\pi}) \\
\lesssim{} & \alpha(C_\alpha^*)^2 + \frac{V_{\max}\sqrt{\log(1/\tau)}}{\tau}\sqrt{\frac{V_{\max}}{\alpha}\left(\frac{\log(|\mathcal{V}|/\delta)}{N}\right)^{1/4} + C_\alpha^*\sqrt{\frac{\log(|\Pi|/\delta)}{N}} + \frac{V_{\max}}{\alpha N}\log(|\Pi|/\delta)} \\
\lesssim{} & \alpha(C_\alpha^*)^2 + \frac{V_{\max}\sqrt{\log(1/\tau)}}{\tau}\sqrt{\frac{V_{\max}C_\alpha^*}{\alpha N^{1/4}}\log(|\mathcal{V}||\Pi|/\delta)} \\
\lesssim{} & \left(\frac{V_{\max}^3(C_\alpha^*)^3\log(1/\tau)\log(|\mathcal{V}||\Pi|/\delta)}{\tau^2 N^{1/4}}\right)^{1/3}
\end{aligned}
$$

where $\alpha \asymp \left(\frac{V_{\max}^3\log(1/\tau)\log(|\mathcal{V}||\Pi|/\delta)}{\tau^2(C_\alpha^*)^3 N^{1/4}}\right)^{1/3}$. $\qquad\square$

## D.2 Stochastic Settings

**Theorem 3** (Statistical rate of the suboptimality in stochastic settings)**.** *Under Assumptions 1 to 5, with probability at least $1 - \delta$, the output policy $\hat{\pi}$ by Algorithm 1 (with the choice of Algorithm 3 for $V$-learning in stochastic settings) satisfies*

$$
J(\pi^*) - J(\hat{\pi}) \lesssim \left(\frac{V_{\max}^3(C_\alpha^*)^3\log(1/\tau)\log(|\mathcal{V}||\mathcal{P}||\Pi|/\delta)}{\tau^2 N^{1/4}}\right)^{1/3}
$$

*if we choose $\alpha \asymp \left(\frac{V_{\max}^3\log(1/\tau)\log(|\mathcal{V}||\mathcal{P}||\Pi|/\delta)}{\tau^2(C_\alpha^*)^3 N^{1/4}}\right)^{1/3}$ and assume $N = N_0$.*

*Proof.* The proof is identical to the proof of Theorem 2 except that we replace $\epsilon_{\text{stat}}$ with $\epsilon_{\text{stat}}^{\text{stochastic}}$. $\qquad\square$

