# OpenReview forum: "Provably Efficient Offline Goal-Conditioned Reinforcement Learning with General Function Approximation and Single-Policy Concentrability"
_NeurIPS.cc/2023/Conference — NeurIPS 2023 poster_

### Official Review · Reviewer_vRJf · 2023-07-04

**Soundness:** 3 good
**Presentation:** 3 good
**Contribution:** 3 good
**Rating:** 6
**Confidence:** 4

**Summary:**

The paper proposes the VP-learning algorithm to solve offline goal-conditioned RL in the context of general function approximations. The algorithm is based on a previous empirically successful algorithm proposed by [1], and the author proves the finite sample complexity for VP-learning under mild assumptions. The proposed  algorithm avoids minimax learning by using a duality formulation first identified by [1], which makes the proposed VP-learning algorithm friendly to implementation while also enjoying theoretical guarantees.

**References:**

[1] Ma J Y, Yan J, Jayaraman D, et al. Offline goal-conditioned reinforcement learning via $ f $-advantage regression[J]. Advances in Neural Information Processing Systems, 2022, 35: 310-323.

**Strengths:**

- The VP-learning algorithm proposed in the paper enjoys finite sample guarantees under a mild single policy concentrability assumption. Compared with previous works on offline RL (single-task) algorithms with general function approximations, VP-learning does not involve minimax optimization, potentially making the algorithm more easy and stable to implement.

**Weaknesses:**

- This theoretical paper aims to study the problem of offline goal-conditioned RL. But it seems that the whole theory does not essentially rely on the setting of goal-conditioned RL. The existing theoretical works this paper compares with are also for standard offline RL, which makes the motivation of the paper confusing. It's true that the standard offline RL can be interpreted as a special case of goal conditioned RL, but I hope to see more intuitions and messages on the problem of goal-conditioned RL itself.
- The key path through which the VP-learning algorithm can avoid solving minimax optimization problems is based on [1], which makes the theoretical contributions and insights of the paper weakened.

**Questions:**

- Regarding Assumption 4 (single policy realizability), what is the $\alpha$ in the equation $\pi_{\alpha}^{\star}\in\Pi$? Since in Theorem 2 and 3 this assumption is imposed before the choice of $\alpha$, the $\alpha$ in the assumption seems undetermined. I think this needs clarification since the paper assumes that the policy class $\Pi$ is finite, which necessitates a specific choice of $\pi_{\alpha}^{\star}$.
- Regarding Assumption 5 (lower bound of policy), this seems a quite strong assumption which is not needed by previous single task offline RL algorithms with general function approximations, e.g., [2, 3, 4]. The author argues that the parameter $\tau$ can be very small, but a small $\tau$ will also increase the upper bound of the suboptimality of VP-learning due to Theorem 2&3. Is this assumption actually  necessary for offline goal-conditioned RL? or is it only required by the specific analysis for VP-learning algorithm?

**References:**

[2] Zhan W, Huang B, Huang A, et al. Offline reinforcement learning with realizability and single-policy concentrability[C]//Conference on Learning Theory. PMLR, 2022: 2730-2775.

[3] Uehara M, Sun W. Pessimistic model-based offline reinforcement learning under partial coverage[J]. arXiv preprint arXiv:2107.06226, 2021.

[4] Xie T, Cheng C A, Jiang N, et al. Bellman-consistent pessimism for offline reinforcement learning[J]. Advances in neural information processing systems, 2021, 34: 6683-6694.

**Limitations:**

The above questions are potential limitations.

---

> ### Author Rebuttal · Authors · 2023-08-09
>
> We thank the reviewer for the helpful and insightful comments. Below are our responses.
>
> >But it seems that the whole theory does not essentially rely on the setting of goal-conditioned RL
>
> We provide the following reasons (motivations) that our paper uses goal-conditioned settings.
>
> 1. The original algorithm GoFAR we analyze is designed for GCRL setting, so we keep their setting. Also, as we compare our VP-learning and GoFAR algorithm empirically (see “global” response for details), the environments are goal-conditioned.
> 2. GCRL is a more general setting that includes single-task RL as a special case, and our algorithm “can also be applied in non-goal-conditioned settings” as Reviewer 39fV mentioned. Since it can be analyzed and applied in both GCRL and single-task settings, we choose the more general one to make our result applicable in a wider scope.
>
>
> >The key path through which the VP-learning algorithm can avoid solving minimax optimization problems is based on [1], which makes the theoretical contributions and insights of the paper weakened.
>
> As Reviewer fB34 said, our algorithm “achieves an effective and reasonable improvement from the lens of theory”, and our theory further provides insightful guidance on the practical implementation of algorithms (as shown in the “global” response and the attached pdf).
>
> We disagree that the above step makes our theoretical contribution and insight weakened. Instead, this is one of the important steps that ensure our algorithm has a theoretical guarantee and outperforms [1] empirically. Note that the GoFAR algorithm in [1] lacks a theoretical guarantee and we provide a finite sample guarantee under a careful choice of $\alpha$. An appropriate value of $\alpha$ not only ensures that our algorithm has a finite sample guarantee (Theorem 1) but also helps to improve over the previous algorithm GoFAR empirically (see “global” response for details).
>
>
> >What is the $\alpha$ in the equation $\pi_\alpha^\star \in \Pi$?
>
> $\alpha$ can be chosen as in Theorem 2 or 3. Our Assumption 4 is actually similar to Assumption 2,3 in [2]. We will mention it in our assumption that $\alpha$ can be chosen as in Theorem 2 or 3. Also, the policy class can have infinite cardinality, and our results still hold as long as the policy class $\Pi$ has a bounded log-covering number. We assume a finite cardinality only for the convenience of presentation (also similar to [2]).
>
> >Regarding Assumption 5 (lower bound of policy), this seems a quite strong assumption which is not needed by previous single task offline RL algorithms with general function approximations, … Is this assumption actually necessary for offline goal-conditioned RL? or is it only required by the specific analysis for VP-learning algorithm?
>
> First, although some previous single tasks offline RL algorithms do not require the lower bound assumption, they require other strong assumptions. For example, [3] requires a completeness-type assumption, where for all policy $\pi \in \Pi$, it requires the $Q$ function of $\pi$  is realized in a value function class $\mathcal{F}$, and $\mathcal{F}$ further needs to satisfy the completeness assumption. [2] does not require a lower bound of policy, but they assume that the behavior policy $\mu$ is known; note that if this is true for our algorithm, then we can directly calculate the policy using $\mu$ and the learned $U_V$ after $V$-learning and thus does not even require a policy class. However, this method is not realistic and thus we use a more practical method (i.e., weighted MLE) to calculate the policy.
>
> Also, we argued that $\tau$ can be very small and even depends on $\alpha$. When $\tau$ depends on $\alpha$, e.g., $\tau = \alpha^c$ for some constant $c > 0$, we can choose a different $\alpha$ between line 588 and 589 in the proof of the main theorem s.t.  the two terms in the third line between line 588 and 589 equal to each other and still obtain a suboptimality rate polynomial in $n$. Roughly speaking, we require that $\alpha = \frac{1}{\alpha^c}\cdot \frac{1}{\alpha^{1/4}N^{1/8}}$, and thus the suboptimality is $O(\alpha) = O(1/N^{\frac{1}{10+8c}})$, which equals to $O(1/N^{1/10})$ when $c = 0$ as in Theorem 2. When $c$ is positive but small, our lower bound assumption on policy class is mild and it only makes the sample complexity slightly worse.
>
> In practice, we don’t need this lower bound assumption and can directly apply the weighted MLE algorithm in the policy learning procedure. We add this assumption for theoretical analysis. To the best of our knowledge, it remains an open question whether a lower bound assumption on policy class is necessary for a finite sample guarantee if one uses MLE.
>
> **Reference:**
>
> [1] Ma J Y, Yan J, Jayaraman D, et al. Offline goal-conditioned reinforcement learning via f-advantage regression[J]. Advances in Neural Information Processing Systems, 2022, 35: 310-323.
>
> [2] Zhan W, Huang B, Huang A, et al. Offline reinforcement learning with realizability and single-policy concentrability[C]//Conference on Learning Theory. PMLR, 2022: 2730-2775.
>
> [3] Xie T, Cheng C A, Jiang N, et al. Bellman-consistent pessimism for offline reinforcement learning[J]. Advances in neural information processing systems, 2021, 34: 6683-6694.

---

> > ### Comment · Reviewer_vRJf · 2023-08-13
> > **Response to Authors**
> >
> > I have read the rebuttal and the comments from other reviewers. Thanks for all your efforts dealing with my concerns. I agree now the theoretical contribution of the paper is enough, especially given that the technical concerns from Reviewer quHd can be addressed. Regarding my other questions, I hope that the authors could make them clearer in the revision. Given all that, I am pleased to raise my score to 6.

---

> > > ### Author Response · Authors · 2023-08-13
> > >
> > > Thanks for your time reviewing our paper, reading our response, and providing helpful feedback! We will make the points you mentioned more clear in the revision.

---

### Official Review · Reviewer_quHd · 2023-07-05

**Soundness:** 3 good
**Presentation:** 3 good
**Contribution:** 3 good
**Rating:** 6
**Confidence:** 4

**Summary:**

This paper provides a rigorous theoretical analysis to a modified version of existing offline Goal-conditioned RL algorithm, and proves that it has an $\tilde O (poly(1/\epsilon))$ sample complexity, where $\epsilon$ is the desired suboptimality of the learned policy. The algorithm requires minimal assumptions on the dataset and the function class and does not involve minimax optimization.

**Strengths:**

The paper makes a novel contribution by providing a theoretical analysis of an existing offline GCRL algorithm.

The paper is clearly written and well-organized. The theoretical analysis is well-explained.


**Weaknesses:**

The derivation of this paper is flawed due to a mistake in Proposition 3.2. This mistake leads to the conclusion that strong duality holds, and we can recover the optimal policy by solving the dual problem. However, it can be seen that, at the limit of $\alpha\to0$, the dual problem (Equation (6)) solves a value function that evaluates the behavior policy. Therefore, solving the dual problem cannot recover the optimal value from suboptimal data, and strong duality does not hold. As a result, all subsequent derivation is invalid, and the claims are unsupported.

The mistake in Proposition 3.2 stems from a citation of Proposition 4.2 of [1] which incorrectly establishes strong duality.

Overall, I think the paper would have been a valuable contribution to the field of offline RL. However, the flawed derivation is a serious issue that needs to be addressed.

[1] Ma, Jason Yecheng, et al. "Offline goal-conditioned reinforcement learning via $f$-advantage regression." Advances in Neural Information Processing Systems 35 (2022): 310-323.


===Post-rebuttal Update===

Previously raised concerns have been effectively addressed. Overall, I am optimistic that, with some revisions, this paper could meet the standards for acceptance. I am pleased to raise my score from 3 to 6 and look forward to seeing your further refinement of this paper.


**Questions:**

Can you explain more on the raised issue?

**Limitations:**

The authors of the paper have discussed the limitations of their work.

---

> ### Author Rebuttal · Authors · 2023-08-09
>
> We thank the reviewer for the helpful and insightful comments. Below are our responses.
>
> >Overall, I think the paper would have been a valuable contribution to the field of offline RL. However, the flawed derivation is a serious issue that needs to be addressed.
>
> We really appreciate the reviewer’s work in identifying the issue that is caused by “a citation of Proposition 4.2 of [1] which incorrectly establishes strong duality”. We carefully checked the derivation of Proposition 4.2 of [1], and found that their result indeed has some issues theoretically. Below we provide a simple fix to this issue with details and show that our theoretical results still hold.
>
> The issue is caused when we solve
>
> $\max\_{d(s,a;g) \geq 0} \mathbb{E}\_{(s,a,g)\sim d} [r(s;g) + \gamma \mathcal{T} V(s,a;g) - V(s;g)]  - D\_g(d(s,a;g) || \mu(s,a;g)).$
>
> For notation convenience, we denote $A_V(s,a;g) = r(s;g) + \gamma  \mathcal{T}  V(s,a;g) - V(s;g)$, and denote $w(s,a;g) = d(s,a;g)/\mu(s,a;g)$. Then the above problem can be rewritten as
>
>  $ {\max}\_{w \geq 0}  \mathbb{E}\_{(s,a,g) \sim \mu}[w(s,a;g) A\_V(s,a;g) - g(w(s,a;g))].$
>
> Note that we can solve $w(s,a;g)$ separately for each individual $(s,a,g)$ pair.
>
> If we don't have the constraint that $w \geq 0$, then equation (22),(23) in [1] is correct, i.e., the maximum of the above object is
>     $\mathbb{E}\_{(s,a,g) \sim \mu}[g\_\*(A\_V(s,a;g))]$
>  and the optimal $w$ satisfies $w\^\star\_V(s,a;g) = g'\_\*(A\_{V}(s,a;g))$. However, we have the constraint that $w \geq 0$, which makes the above solution incorrect. Under this constraint, one can solve that when $g$ is convex, we have
>        $ d\_V^\star(s,a;g) = w\_V^\star (s,a;g) \cdot \mu(s,a;g) = g'\_\*(A\_V(s,a;g))\_+ \cdot \mu(s,a;g)$
>     where $x\_+ \triangleq \max$ {$x, 0$}, and the maximum of the above object is
>
>    $\mathbb{E}\_{(s,a,g)\sim \mu}[I$ { $g'\_\*(A\_V(s,a;g)) \geq 0$ } $\cdot g\_\*(A\_V(s,a;g)) + I$ { $g'\_\*(A\_V(s,a;g)) < 0$ } $\cdot \min_{u \in \mathbb{R}} g\_\*(u)].$
>
> If we further define $\tilde{g}\_\*(x) = g\_\*(x) - \min\_{u \in \mathbb{R}} g\_\*(u)$ which is a constant shift of $g\_\*$, then the above objective can be expressed as
>
>    $ \mathbb{E}\_{(s,a,g)\sim \mu}[ I$ { $g'\_\*(A\_V(s,a;g)) \geq 0$ } $ \cdot \tilde{g}\_\*(A\_V(s,a;g)) ] + \min\_{u \in \mathbb{R}} g\_\*(u).$
>
> Therefore, the $V$-learning should be corrected as $\min_V L_\alpha (V)$ where
>
> $L\_\alpha(V) =  \alpha( (1-\gamma)\mathbb{E}\_{(s,g)\sim(\rho,p(g))}[V(s;g)]  +
>    \mathbb{E}\_{(s,a,g)\sim \mu}[ I$ { $g'\_\*(A\_V(s,a;g)) \geq 0 $ } $ \tilde{g}\_\*(A\_V(s,a;g)) ]).$
>
> Note that this is similar to the previous $L_\alpha(V)$ except that we only consider the $\tilde{g}\_\*$ of which $g'\_\* \geq 0$ in the second term.
>
> Under our choice of $f(x) = \frac{1}{2}(x-1)^2$, $g = \alpha \cdot f$, we have $g_*(x) = \frac{(x+\alpha)^2-\alpha^2}{2\alpha}$. Note that in our paper we defined  $U_V(s,a;g) = r(s;g) + \gamma  \mathcal{T}  V(s,a;g) - V(s;g) + \alpha = A_V(s,a;g) + \alpha$. Therefore, the $V$-learning objective is also equivalent to
>
>   $ L\_\alpha(V) =  \alpha(1-\gamma)\mathbb{E}\_{(s,g)\sim(\rho,p(g))}[V(s;g)] + \frac{1}{2} \mathbb{E}\_{(s,a,g)\sim \mu}[ (U\_V(s,a;g)\_+)^2 ].$
>
> Similarly, in the policy learning procedure, we need to modify the algorithm to
>
>    $ \pi^\*\_\alpha = \arg\max\_{\pi} \mathbb{E}_{(s,a,g)\sim \mu}[(U\_\alpha^*(s,a;g)\_+/\alpha) \cdot \log \pi(a|s,g)].$
>
> Note that this is also similar to the previous form, except that we ignore the term with a negative $U$ value.
>
> With the above modification, our algorithm is correct and the whole proof still goes through. Note that for the most part, the proof remains unchanged, and the only place we need to pay attention to is Lemma 1. Since now our policy learning procedure uses $U_+$ as the weight instead of $U$, we only need to control
> $|| \hat U\_+ - (U^\*\_\alpha)\_+ ||\_{2,\mu}$.
> The previous proof uses the property that $\tilde{L}\_\alpha(U\_V) \triangleq L\_\alpha(V)$ is strongly convex w.r.t. $U\_V$ and $|| \cdot ||_{2,\mu}$. Then we can upper bound $|| \hat U - U^\*\_\alpha ||\_{2,\mu}^2$ using $\tilde{L}\_\alpha(\hat U) -  \tilde{L}\_\alpha(U\_\alpha^*)$.
> After modification, $\tilde{L}\_\alpha(U\_V) \triangleq L\_\alpha(V)$ is no longer strongly convex, but is ``semi''- strongly-convex, i.e., $\tilde{L}\_\alpha(U\_V) - \frac{1}{2}\mathbb{E}\_{(s,a,g)\sim \mu}[ (U\_V(s,a,g)\_+)^2]$ is linear (and thus convex). Therefore, if we define $h(U\_V) = \tilde{L}\_\alpha(U\_V) - \frac{1}{2}\mathbb{E}\_{(s,a,g)\sim \mu}[ (U\_V(s,a,g)\_+)^2]$, we have by the definition of convexity that
>     $h(y) \geq h(x) + \nabla h(x)^{\mathsf{T}} (y-x)$
> which implies that
>    $ \tilde{L}\_\alpha(y) \geq \tilde{L}\_\alpha(x) + \nabla \tilde{L}\_\alpha(x)^{\mathsf{T}}(y-x) + \frac{1}{2}|| y\_+ - x\_+ ||\_{2,\mu}^2.$
> Therefore, we can use the same method to upper bound $|| \hat U\_+ - (U^\*\_\alpha)\_+ ||\_{2,\mu}^2$ using $\tilde{L}\_\alpha(\hat U) -  \tilde{L}\_\alpha(U\_\alpha^\*)$ and all the other proof remains unchanged.
>
> We thank the reviewer again for identifying the issue so that we can fix it and make our theoretical results more solid. We are also encouraged that the reviewer thought our paper would be “a valuable contribution to the field of offline RL” if we could address the issue. We are happy to discuss this further if the reviewer still has any unaddressed concerns.
>
> **Short Version**
>
> $V$-learning should be
>
> $L\_\alpha(V) =  \alpha( (1-\gamma)\mathbb{E}\_{(s,g)\sim(\rho,p(g))}[V(s;g)]  +
>    \mathbb{E}\_{(s,a,g)\sim \mu}[ I$ { $g'\_\*(A\_V(s,a;g)) \geq 0 $ } $ \tilde{g}\_\*(A\_V(s,a;g)) ]).$
>
> and policy learning should be
>
>  $ \pi^\*\_\alpha = \arg\max\_{\pi} \mathbb{E}_{(s,a,g)\sim \mu}[(U\_\alpha^*(s,a;g)\_+/\alpha) \cdot \log \pi(a|s,g)].$
>
> **Reference:**
>
> [1] Ma J Y, Yan J, Jayaraman D, et al. Offline goal-conditioned reinforcement learning via f-advantage regression[J]. Advances in Neural Information Processing Systems, 2022, 35: 310-323.

---

> > ### Comment · Reviewer_quHd · 2023-08-13
> > **I am pleased to raise my score from 3 to 6**
> >
> > I appreciate the effort you've put into addressing the concerns raised in my review. Upon considering your response, I am pleased to acknowledge that the primary concern I had previously expressed has been effectively addressed. However, the proposed fix raises an additional problem. The V-learning now involves a minmax optimization which eliminates a bright spot of the VP-learning algorithm.
> >
> > Furthermore, I am aligned with the viewpoints of Reviewer vRJf concerning the adoption of the GCRL setting. It appears that the inclusion of the GCRL setting introduces complexity to the notation without yielding any benefits. Contrary to the assertion in your rebuttal, I think this setting does not enhance the generality of the approach. The goal can be easily formulated as the additional state dimensions, making GCRL special cases of RL. Therefore, I recommend move to the RL setting which could also amplify the potential impact and reception of this paper.
> >
> > Overall, I am optimistic that, with some revisions, this paper could meet the standards for acceptance. I am pleased to raise my score from 3 to 6 and look forward to seeing your further refinement of this paper.

---

> > > ### Author Response · Authors · 2023-08-13
> > > **Thanks for your valuable feedback!**
> > >
> > > We thank the reviewer again for the helpful comments that make our results more solid, and we are pleased that we have addressed the concern!
> > >
> > > For GCRL vs single-task RL setting, it's a very good suggestion to view GCRL as a special case of RL and simplify the notation to avoid introducing complexity to readers. We will make corresponding modifications in the revision.
> > >
> > > For the additional problem that the reviewer mentioned regarding our proposed fix, actually, the $V$-learning is still a minimization problem. Note that our $V$-learning still has the form of $\min\_{V} L\_\alpha(V)$, and the objective
> > >
> > >   $L\_\alpha(V) =  \alpha( (1-\gamma)\mathbb{E}\_{(s,g)\sim(\rho,p(g))}[V(s;g)]  +
> > >    \mathbb{E}\_{(s,a,g)\sim \mu}[ I$ { $g'\_\*(A\_V(s,a;g)) \geq 0$ } $ \tilde{g}\_*(A\_V(s,a;g)) ])$
> > >
> > >  in our proposed fix does not involve a maximization problem. Therefore, after the fix, our VP-learning algorithm still enjoys the property that it does not involve a minimax optimization.

---

### Official Review · Reviewer_39fV · 2023-07-06

**Soundness:** 3 good
**Presentation:** 4 excellent
**Contribution:** 3 good
**Rating:** 7
**Confidence:** 1

**Summary:**

This paper aims at improving the theoretical understanding of offline goal-conditioned RL (GCRL). In particular, this paper modifies an existing offline GCRL algorithm and shows an O^˜ (poly(1/ϵ)) sample complexity under minimal assumptions of single-policy concentrability and realizability. Their algorithm, called VP-learning, consists of two uninterleaved optimization steps and has good empirical performance while retaining computational stability. Moreover, it can also be applied in non-goal-conditioned settings.

There seems to be a theory-practice gap that this paper addresses. Namely, most provably efficient algorithms seems to require minimax optimization, while in practice that is not effective. Ideally, an algorithm should be practical with good sample complexity guarantees, which is a gap this paper aims to fill.

In particular this paper provides guarantees for a modified version of an algorithm GoFAR, with modifications in the deterministic and stochastic MDP settings. They call their modified algorithm VP-learning. While other algorithms have been shown to be efficient under single-policy concentrability and realizability assumptions, they require solving minimax optimization problems.


**Strengths:**

While other algorithms have been shown to be efficient under single-policy concentrability and realizability assumptions, they require solving minimax optimization problems. This algorithm does not.

The algorithm they develop is built off of an algorithm with good empirical performance.


**Weaknesses:**

While part of the pitch of the paper is that they desire an algorithm that has provable efficiency whilst also having good empirical performance, they do not have empirical results to demonstrate that their modified algorithm performs well.


**Questions:**

How do you expect the modified algorithm, VP-learning to compare to GoFAR in terms of empirical performance? How do you expect the modifications will impact things adversely?


**Limitations:**

To my knowledge, the authors do adequately address the limitations of the paper.

---

> ### Author Rebuttal · Authors · 2023-08-09
>
> We thank the reviewer for the helpful and insightful comments. Below are our responses.
>
> >How do you expect the modified algorithm, VP-learning to compare to GoFAR in terms of empirical performance? How do you expect the modifications will impact things adversely?
>
> The modified algorithm, VP-learning outperforms GoFAR in terms of empirical performance, and we provided details in the “global” response.

---

> > ### Comment · Reviewer_39fV · 2023-08-18
> > **Response to Authors**
> >
> > Thank you for answering my question.

---

> > > ### Author Response · Authors · 2023-08-18
> > >
> > > Thanks again for your time and effort in reviewing our paper and reading our response.

---

### Official Review · Reviewer_fB34 · 2023-07-06

**Soundness:** 3 good
**Presentation:** 3 good
**Contribution:** 3 good
**Rating:** 6
**Confidence:** 2

**Summary:**

This paper establishes a rigorous theoretical analysis for offline goal-conditioned reinforcement learning algorithms (GCRL). To achieve that, the authors made a slight modification on top of an existing offline GCRL algorithm (GoFAR), achieve a polynomial sample complexity by regression instead minimax optimization.

**Strengths:**

1. This paper is well-organized and provides a throughout survey of related work.
2. It seems to achieve an effective and reasonable improvement from the lens of theory.

**Weaknesses:**

There is no experiment to support the correctness of the theoretical analysis.

**Questions:**

N/A

---

> ### Author Rebuttal · Authors · 2023-08-09
>
> We thank the reviewer for the helpful and insightful comments. Below are our responses.
>
> >There is no experiment to support the correctness of the theoretical analysis.
>
> We provide experiments to support the correctness of our theoretical analysis (especially the choice of $\alpha$). Please see the “global” response for details.

---

> > ### Comment · Reviewer_fB34 · 2023-08-21
> > **Thanks for your response**
> >
> > I thank the authors for their response and the uploaded experiment results, which improve the credibility of this work. I do not doubt the correctness of their contribution to the theory, so I gave such a suggestion as I think it would be helpful for the readers to have a straightforward understanding of their work. I'll improve my score from 5 to 6. However, I still have some suggestions about their uploaded experiment results: I cannot get any information about the environment settings, e.g., what are FetchReach, FetchPick, FetchPush, FetchSlide, and HandReach? I hope the authors can add this part of the environment introduction and also the learning curve in their future revision.

---

> > > ### Author Response · Authors · 2023-08-21
> > > **Thanks for your suggestion**
> > >
> > > Thanks for your time reviewing our paper and reading our response, and thanks for raising your score! For the experiment settings, FetchReach, FetchPick, FetchPush, FetchSlide, and HandReach are all environments in the d4rl benchmark. We did not provide details of the environment in the global response since it is the same as [1], and thus we omit the details to keep the response more concise. We will add the environmental details in the revision. Thanks for your suggestion!
> > >
> > >
> > > **Reference:**
> > >
> > > [1] Ma J Y, Yan J, Jayaraman D, et al. Offline goal-conditioned reinforcement learning via f-advantage regression[J]. Advances in Neural Information Processing Systems, 2022, 35: 310-323.

---

### Author Rebuttal · Authors · 2023-08-09

We thank all the reviewers for their helpful and insightful comments. Below we first address common issues.  Since several reviewers mentioned that our paper does not provide empirical results of the modified algorithm, we provide experimental results of our VP-learning algorithm with different choices of $\alpha$ under five environments (FetchReach, FetchPick, FetchPush, FetchSlide, and HandReach) used in [1] (see the two tables in the attached pdf file ). All the implementation details of our VP-learning are the same as the GoFAR algorithm [1], except for the value of $\alpha$. Note that our VP-learning algorithm with $\alpha=1$ is equivalent to the GoFAR algorithm. Table 1 contains the discounted returns and Table 2 contains the final distances of the policies trained after 100 epochs and evaluated over 10 runs. For each environment and each $\alpha$, the result was averaged over 3 random seeds as in the GoFAR paper [1]. The best results of each environment are in bold.

The empirical results demonstrate the correctness of our theoretical analysis: choosing $\alpha=1$ will result in a large suboptimality of $\pi_\alpha^\star$ and thus the learned policy $\hat \pi$. Instead, we should carefully choose the value of $\alpha$ to ensure a vanishing suboptimality. In practice, we should tune the value of $\alpha$ and typically it should be less than 1. In our experiments, the best $\alpha$ ranges over $0.05-0.5$.

**Reference:**

[1] Ma J Y, Yan J, Jayaraman D, et al. Offline goal-conditioned reinforcement learning via f-advantage regression[J]. Advances in Neural Information Processing Systems, 2022, 35: 310-323.

---

### Decision · Program_Chairs · 2023-09-21

**Decision:**

Accept (poster)

**Comment:**

This paper provides a new theoretical understanding of offline goal-conditioned RL (GCRL). In particular, under the assumptions of single-policy concentrability and realizability, this paper modifies an existing offline GCRL algorithm and shows an $\tilde{O}(Ploy(1/\epsilon))$ sample complexity via VP-learning. I recommend the paper be accepted. The authors should incorporate the reviewers suggestions in the revision. In addition, there are two missing papers that I believe is relevant: [1] initiated the theoretical study for the offline stochastic shortest path problem, albeit for the tabular setting; Beyond  Rashidinejad et al., 2021, [2] considered the offline reinforcement learning with single-policy concentrability and achieved instance-depend characterization. It is of great importance to ask (beyond the worst-case guarantee) whether instance-depend bound can be arrived for offline goal-conditioned learning. The authors should include the discussion for [1],[2] in the camera-ready.

[1] Offline Stochastic Shortest Path: Learning, Evaluation and Towards Optimality, UAI22
[2] Towards instance-optimal offline reinforcement learning with pessimism, NeurIPS21